# MMDisCo: Multi-Modal Discriminator-Guided Cooperative Diffusion for Joint Audio and Video Generation

**Akio Hayakawa**[1]     **Masato Ishii**[1]     **Takashi Shibuya**[1]     **Yuki Mitsufuji**[1,2]
[1]Sony AI     [2]Sony Group Corporation
{akio.hayakawa,masato.a.ishii,takashi.tak.shibuya,yuhki.mitsufuji}@sony.com

## Abstract

This study aims to construct an audio-video generative model with minimal computational cost by leveraging pre-trained single-modal generative models for audio and video. To achieve this, we propose a novel method that guides single-modal models to cooperatively generate well-aligned samples across modalities. Specifically, given two pre-trained base diffusion models, we train a lightweight joint guidance module to adjust scores separately estimated by the base models to match the score of joint distribution over audio and video. We show that this guidance can be computed using the gradient of the optimal discriminator, which distinguishes real audio-video pairs from fake ones independently generated by the base models. Based on this analysis, we construct a joint guidance module by training this discriminator. Additionally, we adopt a loss function to stabilize the discriminator's gradient and make it work as a noise estimator, as in standard diffusion models. Empirical evaluations on several benchmark datasets demonstrate that our method improves both single-modal fidelity and multimodal alignment with relatively few parameters. The code is available at: https://github.com/SonyResearch/MMDisCo.

## 1 Introduction

Deep generative modeling has progressed rapidly in the last few years. Diffusion models are one of the keys to this progress, and they can be applied to various tasks, including image, audio, and video generation (Yang et al., 2023). Following the success of single-modal data, several attempts have been made to apply diffusion models to multimodal data (Bao et al., 2023). However, since multimodal data is more complex and harder to collect than single-modal data, developing multimodal generative models by simply extending single-modal ones remains challenging. One promising way to alleviate this problem is to integrate several pre-trained single-modal models to build a multimodal generative model (Tang et al., 2023; Xing et al., 2024). As there are numerous publicly-available models that can generate high-quality single-modal data (Rombach et al., 2022; Liu et al., 2023; Guo et al., 2024), their effective integration would substantially reduce the computational cost to build multimodal generative models. In this work, we focus on audio-video joint generation on top of two pre-trained diffusion models for audio and video.

There are two approaches for audio-video generation that integrate several single-modal models: training-free and training-based. The training-free approach employs pre-trained single-modal base generative models with their parameters fixed. It uses an off-the-shelf recognition model to guide them to generate well-aligned samples across modalities (Xing et al., 2024). While this can improve multimodal alignment without any training cost, it may degrade the fidelity of a single modality (Xing et al., 2024). In contrast, the training-based approach extends single-modal generative models for multimodal data by designing a neural network tailored to it (Ruan et al., 2023; Tang et al., 2023). Although this approach can achieve better performance in terms of both multimodal alignment and the fidelity of each single modality, it tends to require a significant computational cost for training. More importantly, their architectures for handling multimodal data heavily depend on those of base models (i.e., they are not model-agnostic). Therefore, when updating base models, we must manually redesign them, which requires a lot of trial and error. In short, the existing two approaches involve

a trade-off between the quality of generated samples and model dependency, which increases the computational cost.

In this paper, we propose a novel method that is training-based but model-agnostic. Our method does not require backpropagation through the base models for the optimization. Specifically, we introduce a lightweight joint guidance module on top of audio and video base models that adjust their outputs for audio-video joint generation. We assume that pre-trained base models are black box diffusion models (i.e., we can access only their outputs and do not depend on a specific architecture design like a cross-attention module to construct a joint generation model). We formulate the joint generation process as an extension of the classifier guidance (C-guide) for single-modal data (Song et al., 2021; Dhariwal and Nichol, 2021). We show that this joint guidance can be computed through the gradient of the optimal discriminator that distinguishes real audio-video pairs from the fake ones independently generated by base models. We only train the discriminator with proper regularization inspired by Denoising Likelihood Score Matching (DLSM) (Chao et al., 2022). Extensive experiments on several benchmark datasets demonstrate that our proposed method can efficiently integrate single-modal base models for audio and video into a joint generation model, maintaining the performance of each single-modal generation without incurring a significant computational cost (see Appendix A.9).

## 2 RELATED WORK

### 2.1 AUDIO-VIDEO JOINT GENERATION BY DIFFUSION MODELS

Since an audio-video pair is one of the most popular types of multimodal data, several works train diffusion models with such pairs to achieve a conditional single-modal generation: video-conditional audio generation (Luo et al., 2023; Mo et al., 2023; Su et al., 2023; Pascual et al., 2024) or audio-conditional video generation (Jeong et al., 2023; Lee et al., 2023; Yariv et al., 2024). However, these works mainly focus on a single modality as a generation target. Extending these works to the joint generation of audio and video is not trivial due to the high dimensionality and heterogeneous data structure of audio-video joint data.

Joint generation of audio and video pairs has been addressed in only a few recent studies (Ruan et al., 2023; Tang et al., 2023; Xing et al., 2024). MM-Diffusion (Ruan et al., 2023) is a multimodal diffusion model specific for audio-video joint generation. While MM-Diffusion is trained from scratch on audio-video pairs, CoDi (Tang et al., 2023) integrates several pre-trained single-modal diffusion models by adopting environment encoders to share modality-specific information across modalities during the generation process. Since they adopt a novel architecture strongly tied to the main network of diffusion models, it is difficult to directly apply their method to other architectures, hindering its applicability. In contrast, our method handles base models as black boxes and depends only on their outputs. Therefore, our method is widely applicable to any type of architecture used in base models.

Xing et al. (2024) shares a similar motivation to ours in the sense that they achieve audio-video cooperative generation from pre-trained single-modal base models. Given multimodal embedding models (e.g., ImageBind (Girdhar et al., 2023)), they utilize universal guidance (Bansal et al., 2023) to make the embeddings from two modalities close. Although their approach is model-agnostic and can be applied to any base model, their guidance roughly ensures semantic alignment in the space of embeddings learned by ImageBind. Thus, it does not achieve sampling from the actual joint distribution of audio-video pairs. In contrast, our method is theoretically grounded in adjusting the scores predicted from base models to the score of the joint distribution, explicitly achieving sampling from the joint distribution.

### 2.2 GUIDANCE FOR PRE-TRAINED DIFFUSION MODELS

The guidance (Song et al., 2021; Dhariwal and Nichol, 2021) provides a proper way to update intermediate representations at each generation step so that the generated samples satisfy a given condition, even when the model is not trained for that type of conditional generation. Since the guidance does not require additional training in diffusion models, it is widely used to control the generation process with additional conditional signals. C-guide (Song et al., 2021; Dhariwal and Nichol, 2021) was proposed to guide an image generation model by the class label, such as *"cat"* or *"dog"*, using an additionally trained classifier. Several works extend this to utilize off-the-shelf

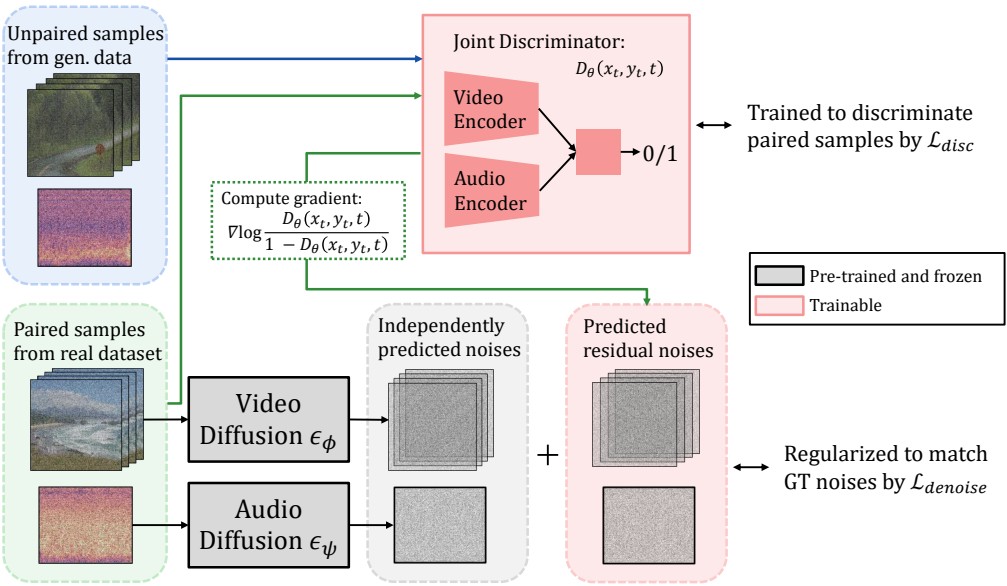

**Figure 1:** Overview of the training process of our proposed method. We train a joint discriminator on the top of two base diffusion models to distinguish real video-audio pairs from fake ones generated by base models. Additionally, we adopt a denoising objective, as in standard diffusion models, to match the gradient of the discriminator with regard to the inputs to the residual noise between ground truth noises and predicted noises from base models.

recognition models to achieve beyond class-conditional generation (Graikos et al., 2022; Bansal et al., 2023). Our proposed method is the first attempt to extend the C-guide for joint multimodal generation. We derive our methodology from the theory of C-guide, enabling us to sample data from the joint distribution.

The samples generated with C-guide may suffer from degraded quality without careful tuning of its scale (Dhariwal and Nichol, 2021). Chao et al. (2022) denote this problem as the score mismatch issue, where the posterior scores estimated by a diffusion model and a classifier are unstable and deviate from the true ones. They proposed DLSM loss to alleviate this problem. DLSM regularizes a classifier's gradient to match the residual error of noise prediction by trained diffusion models, providing a stable gradient for the generation process. Our work can be seen as an extension of DLSM for multimodal generation. We employ DLSM to stabilize the gradient of our discriminator.

Kim et al. (2023) proposed Discriminator Guidance, which guides a Text-to-Image (T2I) model by a discriminator distinguishing real images from generated images to improve the quality of the generated samples by a pre-trained T2I model. Although Kim et al. (2023) share a similar concept to ours regarding using a discriminator to bridge the gap between the score predicted by a pre-trained model and the target score, our goal is to integrate single-modal models into a joint generation model. Their method is particularly designed to handle the gap within a single modality, and it cannot be straightforwardly applied to the multimodal generation. In contrast, our method is derived from directly bridging the gap between a single-modal distribution and a joint one. Moreover, we show a single discriminator can serve as a guidance module for both domains, resulting in minimal computational cost for developing a joint generation model.

## 3 METHODOLOGY

In this section, we first briefly review Diffusion models on a single modal data. Here, we follow Denoising Diffusion Probabilistic Models (DDPM) (Ho et al., 2020), broadly used as a standard definition of diffusion models. Then, we describe our formulation of joint score estimation on top of two pre-trained diffusion models.

### 3.1 PRELIMINARY: DIFFUSION MODELS

**Basics** Diffusion models are a family of probabilistic generative models that reverse a diffusion process from data to pure noise. Specifically, let $x_0 \sim p(x_0)$ be a sample of a data distribution and $x_t$ be a noisy representation at diffusion timestep $t \in \{0, 1, ..., T\}$. A forward diffusion process is defined as a Markov process:

$$q(x_t|x_{t-1}) = \mathcal{N}(\sqrt{1 - \beta_t}x_{t-1}, \beta_t I), \tag{1}$$

where $\beta_t \in (0, 1)$ controls how fast the data is diffused at each timestep. On the basis of Eq. (1), $x_t|x_0$ also follows the Gaussian distribution of $p(x_t|x_0) = \mathcal{N}(\sqrt{\bar{\alpha}_t}x_0, (1 - \bar{\alpha}_t)I)$, where $\bar{\alpha}_t = \prod_{s=1}^{t}(1 - \beta_s)$ is the accumulation of diffusion coefficients, and the noisy sample at the last step $x_T$ would follow the standard Gaussian distribution $\mathcal{N}(0|I)$ with an appropriate setting of $T$ and $\beta_t$. If $\beta_t$ is small enough (or $T$ is large enough), the reverse process of Eq. (1) can be approximated to be Gaussian. Diffusion models are trained to estimate its mean by predicting the noise in $x_t$ as follows:

$$p_\phi(x_{t-1}|x_t) = \mathcal{N}(\mu_\phi(x_t, t), \sigma_t^2 I), \tag{2}$$

$$\mu_\phi(x_t, t) = \frac{1}{\sqrt{1 - \beta_t}}\left(x_t - \frac{\beta_t}{\sqrt{1 - \bar{\alpha}_t}}\epsilon_\phi(x_t, t)\right), \ \sigma_t^2 = \frac{1 - \bar{\alpha}_{t-1}}{1 - \bar{\alpha}_t}\beta_t, \tag{3}$$

where $\epsilon_\phi$ represents the noise prediction model with parameters $\phi$. This model $\epsilon_\phi$ can be trained by minimizing the following mean squared error between the noise predicted by the model and that added to the data:

$$\phi^* = \underset{\phi}{\arg\min} \ \mathbb{E}_{x_0, \epsilon, t} \left\| \epsilon_\phi(\sqrt{\bar{\alpha}_t}x_0 + \sqrt{1 - \bar{\alpha}_t}\epsilon, t) - \epsilon \right\|^2, \tag{4}$$

where $\epsilon \sim \mathcal{N}(0, I)$ is a noise, and $t \sim \mathcal{U}(1, T)$ is a timestep. For generation, we can sample $x_0$ through the iterative sampling from $t = T$ to 1 using Eq. (2) with $\epsilon_{\phi^*}$.

Equation (4) is a form of denoising score matching. Specifically, from Tweedie's formula (Efron, 2011), the noise added to the data and the score function is equivalent up to a constant factor. On the basis of this fact, we can approximate the score function by using a noise prediction network trained by Eq. (4) as:

$$\nabla_{x_t} \log q(x_t) \approx -\frac{1}{\sqrt{1 - \bar{\alpha}_t}}\epsilon_{\phi^*}(x_t, t). \tag{5}$$

From Eq. (5), a noise prediction model trained by Eq. (4) can be considered as a model estimating score of $q(x_t)$.

**Guidance for conditional generation** We can extend the generation process of diffusion models to the conditional one using classifier guidance (C-guide) (Song et al., 2021; Dhariwal and Nichol, 2021). The C-guide can be derived from the perspective of score estimation. By Bayes' theorem, we can write the conditional score as:

$$\nabla_{x_t} \log q(x_t|c) = \nabla_{x_t} \log q(x_t) + \nabla_{x_t} \log q(c|x_t), \tag{6}$$

where $c$ is a conditional vector. The first term on the right-hand side can be estimated by $\epsilon_{\phi^*}$ trained with Eq. (4). The second term on the right-hand side can be estimated by computing a classifier's gradient with respect to its input, where the classifier is trained to predict $c$ given $x_t$.

### 3.2 JOINT SCORE ESTIMATION ON THE TOP OF PRE-TRAINED DIFFUSION MODELS

We aim to achieve joint generation based on two independently trained diffusion models. Namely, let $x \in \mathbb{R}^{D_x}$ and $y \in \mathbb{R}^{D_y}$ be samples of two different modalities. We want to sample $x$ and $y$

from a joint distribution $q(x, y)$. From the perspective of the score function, we need to estimate $\nabla_{x_t} \log q(x_t, y_t)$ and $\nabla_{y_t} \log q(x_t, y_t)$. By Bayes rule, we can derive the following equations:

$$\nabla_{x_t} \log q(x_t, y_t) = \nabla_{x_t} \log q(x_t) + \nabla_{x_t} \log q(y_t|x_t), \tag{7}$$

$$\nabla_{y_t} \log q(x_t, y_t) = \nabla_{y_t} \log q(y_t) + \nabla_{y_t} \log q(x_t|y_t). \tag{8}$$

Similar to the classifier guidance, the first terms in the right-hand side of Eqs. (7) and (8) can be estimated by the two diffusion models independently trained on the modalities $x$ and $y$. On the other hand, modeling the second terms on the right-hand side are not trivial. One can naively construct two additional generative models of $x_t|y_t$ and $y_t|x_t$ and compute the gradient of such models with regard to condition vectors. However, training them is difficult due to high dimensionality and requires a significant computational cost. Instead of training additional high-cost generative models, we train a single lightweight discriminator that distinguishes real pairs of $(x_t, y_t)$ from fake ones generated by pre-trained single-modal base models. We theoretically show that it is sufficient to compute the gradient of this discriminator to approximate these terms.

Specifically, we propose to train a discriminator between the joint distribution $q(x_t, y_t)$ and independent distribution $q(x_t)q(y_t)$. Let $x' \sim p_\phi(x)$ and $y' \sim p_\psi(y)$ be fake paired samples independently generated by pre-trained diffusion models. Here, we denote these diffusion models independently pre-trained by each modality $x$ and $y$ as $\epsilon_\phi^{(x)}$ and $\epsilon_\psi^{(y)}$, where $\phi$ and $\psi$ are the parameters of these models. We train a discriminator $D_\theta : x_t, y_t, t \rightarrow [0, 1]$ with the following loss function:

$$\theta^* = \underset{\theta}{\operatorname{argmin}} \, \mathbb{E}_{(x,y) \sim q(x,y), x' \sim p_\phi(x), y' \sim p_\psi(y)} \left[ \mathcal{L}_{\text{disc}}^{(\theta)}(x, y, x', y') \right], \tag{9}$$

$$\mathcal{L}_{\text{disc}}^{(\theta)}(x, y, x', y') = \mathbb{E}_{\epsilon^{(x)}, \epsilon^{(y)}, \epsilon^{(x')}, \epsilon^{(y')}, t} \left[ \log D_\theta(x'_t, y'_t, t) + \log \left( 1 - D_\theta(x_t, y_t, t) \right) \right], \tag{10}$$

where $\epsilon^{(z)}$ and $z_t$ are a noise and a noisy sample for $z \in \{x, y, x', y'\}$, respectively. The noisy sample $z_t$ is derived by a forward process $z_t = \sqrt{\bar{\alpha}_t^z} z + \sqrt{1 - \bar{\alpha}_t^z} \epsilon^{(z)}$, where $\bar{\alpha}_t^z$ is a coefficient of the forward diffusion process for each modality, and this is omitted from Eq. (10) for brevity. Note that this discriminator $D_\theta$ is trained to output one for real pairs and zero for fake pairs.

Similar to a discriminator in Generative Adversarial Networks (GANs) (Goodfellow et al., 2014), an optimal discriminator $D_{\theta^*}(x_t, y_t, t)$ that minimizes Eq. (10) can be seen as an estimator of the density ratio $\frac{q(x_t, y_t)}{q(x_t, y_t) + p_\phi(x_t)p_\psi(y_t)}$. Therefore, the second term in the right-hand side of Eqs. (7) and (8) can be approximated by utilizing $D_{\theta^*}$ as follows (see Appendix A.1 for details of this derivation):

$$\nabla_{x_t} \log q(y_t|x_t) \approx \nabla_{x_t} \log \frac{D_{\theta^*}(x_t, y_t, t)}{1 - D_{\theta^*}(x_t, y_t, t)}, \tag{11}$$

$$\nabla_{y_t} \log q(x_t|y_t) \approx \nabla_{y_t} \log \frac{D_{\theta^*}(x_t, y_t, t)}{1 - D_{\theta^*}(x_t, y_t, t)}. \tag{12}$$

In summary, we can estimate a joint score as the sum of the scores independently estimated by base models $\epsilon_\phi^{(x)}$ and $\epsilon_\psi^{(y)}$, and the gradient of an optimal discriminator $D^*$ shown in Eqs. (11) and (12).[1] In inference, we independently compute the outputs from base models and the gradient of our discriminator for the intermediate noisy samples $x_t$ and $y_t$ at each timestep. Then, their sum is used as a predicted noise for the denoising step. Note that, in the discussion above, we assume that the distribution of samples generated by base models is equal to the marginal distribution of joint data for brevity (i.e., $q(x_t) = p_\phi(x_t)$ and $q(y_t) = p_\phi(y_t)$). However, our proposed method can be applied even when these are not equal (see Appendix A.2 in the appendix for more details). We also tested this setting in our experiment.

---

[1] In this work, we adopt the most basic form of density ratio estimator. We expect that using advanced ones may improve the performance of joint guidance, which we leave for future work.

| **Algorithm 1** Training process of $D_\theta$. | **Algorithm 2** Joint inference by $D_{\theta*}$. |
|---|---|
| **Require:** $\epsilon_\phi^{(x)}$, $\epsilon_\psi^{(y)}$, and paired dataset
  **repeat**
    Sample $x, y$ from paired dataset.
    Generate $x'$ using $\epsilon_\phi^{(x)}$.
    Generate $y'$ using $\epsilon_\psi^{(y)}$.
    Compute $\mathcal{L}_{\text{disc}}(x, y, x', y')$ by Eq. (10).
    Compute $\mathcal{L}_{\text{denoise}}(x, y)$ by Eq. (13).
    Update $\theta$ based on $\nabla_\theta \mathcal{L}_{\text{all}}$.
  **until** converged
  **Return** $D_{\theta*}$. | **Require:** $\epsilon_\phi^{(x)}$, $\epsilon_\psi^{(y)}$, and $D_{\theta*}$
  Initialize $x_T, y_T$ with Gaussian noise.
  **for** $t$ in $[T, ..., 1]$ **do**
    $D_{\text{ratio}} \leftarrow \log \frac{D_{\theta*}(x_t, y_t, t)}{1 - D_{\theta*}(x_t, y_t, t)}$.
    $\hat{\epsilon}^{(x)} \leftarrow \epsilon_\phi^{(x)}(x_t, t) - \sqrt{1 - \bar{\alpha}_t^x} \nabla_x D_{\text{ratio}}$.
    $\hat{\epsilon}^{(y)} \leftarrow \epsilon_\psi^{(y)}(y_t, t) - \sqrt{1 - \bar{\alpha}_t^y} \nabla_y D_{\text{ratio}}$.
    Sample $(x_{t-1}, y_{t-1})$ based on $(\hat{\epsilon}^{(x)}, \hat{\epsilon}^{(y)})$.
  **end for**
  **Return** $(x_0, y_0)$. |

## 3.3 RESIDUAL SCORE ESTIMATION FROM AN OPTIMAL DISCRIMINATOR

As we described in Section 3.2, we can sample $x$ and $y$ from a joint distribution using guidance with a discriminator that can distinguish real paired data from paired samples generated by pre-trained single-modal diffusion models. However, in our preliminary experiments, using a discriminator trained by only Eq. (9) degrades the fidelity of generated samples as a single modality. We conjectured that this is caused by the score estimation mismatch issue mentioned by Chao et al. (2022). They argued that the score may deviate from the true one when estimating it using a gradient of a classification model. To alleviate this issue, inspired by Chao et al. (2022), we adopt regularization for the gradient of the discriminator to match the residual of true noises and noises predicted by the base diffusion models. Specifically, we define a denoising regularization loss as follows:

$$\mathcal{L}_{\text{denoise}}^{(\theta, \phi, \psi)}(x, y) = \mathbb{E}_{\epsilon^{(x)}, \epsilon^{(y)}, t} \left[ \mathcal{L}_{\text{denoise}}^{(\theta, \phi)} \left( x, y, \epsilon^{(x)}, \epsilon^{(y)}, t \right) + \mathcal{L}_{\text{denoise}}^{(\theta, \psi)} \left( x, y, \epsilon^{(x)}, \epsilon^{(y)}, t \right) \right], \quad (13)$$

$$\mathcal{L}_{\text{denoise}}^{(\theta, \phi)} \left( x, y, \epsilon^{(x)}, \epsilon^{(y)}, t \right) = \left\| \epsilon^{(x)} - \epsilon_\phi^{(x)}(x_t, t) + \sqrt{1 - \bar{\alpha}_t^x} \nabla_{x_t} \log \frac{D_\theta(x_t, y_t, t)}{1 - D_\theta(x_t, y_t, t)} \right\|^2, \quad (14)$$

$$\mathcal{L}_{\text{denoise}}^{(\theta, \psi)} \left( x, y, \epsilon^{(x)}, \epsilon^{(y)}, t \right) = \left\| \epsilon^{(y)} - \epsilon_\psi^{(y)}(y_t, t) + \sqrt{1 - \bar{\alpha}_t^y} \nabla_{y_t} \log \frac{D_\theta(x_t, y_t, t)}{1 - D_\theta(x_t, y_t, t)} \right\|^2. \quad (15)$$

Note that the base models' parameters $\phi, \psi$ are fixed during training and just used to compute noise estimation errors. Our final objective (denoted $\mathcal{L}_{\text{all}}$ hereinafter) to train the discriminator $D_\theta$ is the sum of the discriminator loss (Eq. (10)) and denoising regularization loss (Eq. (13)):

$$\theta^* = \underset{\theta}{\arg\min} \, \mathbb{E}_{(x,y) \sim q(x,y), x' \sim p_\phi(x), y' \sim p_\psi(y)} \left[ \mathcal{L}_{\text{disc}}^{(\theta)}(x, y, x', y') + \lambda \mathcal{L}_{\text{denoise}}^{(\theta, \phi, \psi)}(x, y) \right], \quad (16)$$

where $\lambda$ is a weight to balance these two losses. We use $\lambda = 1$ throughout this paper. The training and generation process of our proposed method is summarized in Algorithms 1 and 2, respectively.

## 4 EXPERIMENTS

In this section, we first show the results of preliminary experiments with toy datasets to confirm our proposed method can guide the generation process of two independently trained diffusion models to the desired joint distribution. Then, we show the experimental results with real data.

### 4.1 PRELIMINARY EXPERIMENTS WITH TOY DATASETS

**Datasets** To create the training dataset for a base model, we evenly sampled data from five Gaussian distributions whose means were $[-3, -1.5, 0, 1.5, 3]$, and variances were all $0.01$, resulting in 500 samples. For training the guidance module, we constructed two types of

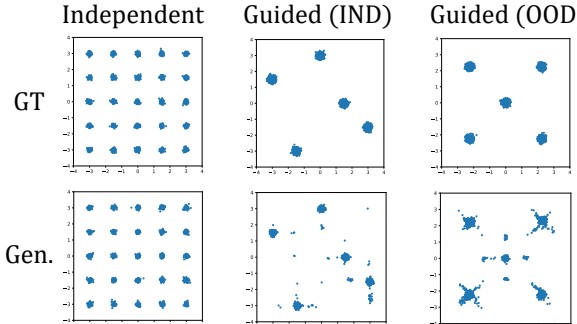

Independent    Guided (IND)    Guided (OOD)

GT

Gen.

**Figure 2:** Visualization of guidance results on toy datasets. The top row shows samples drawn from ground truth distribution (GT), and the bottom row shows generated samples (Gen.).

**Table 1:** Negative log likelihood (NLL) of generated samples over target distribution.

| Dataset | Method | NLL $\downarrow$ |
|---|---|---|
| IND | GT samples | 1.67 |
| | No joint | 18.80 |
| | $\mathcal{L}_{\mathrm{disc}}$ | 3.29 |
| | $\mathcal{L}_{\mathrm{denoise}}$ | 2.40 |
| | $\mathcal{L}_{\mathrm{all}}$ | **1.94** |
| OOD | GT samples | 1.67 |
| | No Joint | 22.10 |
| | $\mathcal{L}_{\mathrm{disc}}$ | 16.60 |
| | $\mathcal{L}_{\mathrm{denoise}}$ | 2.63 |
| | $\mathcal{L}_{\mathrm{all}}$ | **2.53** |

datasets. The first one simulates an in-domain (IND) situation, where the marginal distributions $q(x)$ and $q(y)$ were equal to the distribution on which the base model had been trained. This dataset was constructed by sampling from five Gaussian distributions whose means were $[(-3, 1.5), (-1.5, -3), (0, 3), (1.5, 0), (3, -1.5)]$, respectively. The second one simulates an out-of-domain (OOD) situation, where the marginal distributions differ from the distribution on which the base model is trained. This dataset was constructed by sampling from five Gaussian distributions whose means were $[(-2.25, -2.25), (2.25, 2.25), (-2.25, 2.25), (2.25, -2.25), (0, 0)]$, respectively. We set the same variance for all Gaussians as 0.01. In the former case, the peaks of the distribution are a subset of the independent data distribution, and its marginal distributions for $x$ and $y$ are the same as the data used to train the base model. On the other hand, the peaks are shifted in the latter case, and the marginal distribution differs from the original. To train our discriminator by $\mathcal{L}_{\mathrm{disc}}$ (Eq. (9)), we sampled 500 fake pairs in advance from the base model.

**Setup** We first trained a base diffusion model on scalar value samples from a mixture of five Gaussian distributions. The base diffusion model outputs a single scalar $x \in \mathbb{R}$, and we duplicated the trained base model to generate a two-dimensional vector $(x, y) \in \mathbb{R}^2$ by concatenating outputs from them. Since each $x$ and $y$ has five peaks as a distribution, $(x, y)$ has 25 peaks when $x$ and $y$ are generated independently, as shown in the leftmost column in Fig. 2. Then, we guided the base models using our proposed method to generate samples that follow the target joint distribution. We used a neural network consisting of five fully connected layers for the base diffusion and one with three layers for the guidance module. See Appendix A.3 for more details about the settings.

**Results** Figure 2 shows the generated samples from our proposed method. Our proposed model successfully guides the base models to generate samples from the target distributions in in- and out-of-domain settings. For an ablation study, we evaluated the effectiveness of each loss by negative log-likelihood (NLL) against the target distribution (Table 1). We confirmed that both losses, $\mathcal{L}_{\mathrm{disc}}$ and $\mathcal{L}_{\mathrm{denoise}}$, contribute to substantially improving NLL. These results demonstrate that our proposed method effectively guides base models in generating samples of the target joint distribution.

### 4.2 EXPERIMENTS WITH BENCHMARK DATASETS UNDER THE IN-DOMAIN SETTING

**Setup** To investigate the applicability of our proposed method to real data, we first applied it to an in-domain setting. In this setting, we used the same dataset to train the base models and our proposed model. We trained our discriminator on top of MM-Diffusion (Ruan et al., 2023), which was already trained to generate audio-video data jointly, and guided it to generate further aligned samples. This setting is similar to adopting guidance to a conditional diffusion model with a condition that the diffusion model can directly handle. The discriminator consists of a stack of two-stream ResBlock layers for each audio and video, followed by a linear layer to output a scalar. The total number of its parameters is 12.7M, whereas MM-Diffusion has 133M parameters. We used the pre-trained MM-Diffusion released in the official repository and only trained our discriminator while freezing

**Table 2:** Quantitative evaluation on Landscape and AIST++ datasets for the in-domain adaptation setting. We use MM-Diffusion trained on Landscape or AIST++ dataset at a $64 \times 64$ resolution and additionally guide its outputs by our proposed method.

| Dataset | Method | FVD $\downarrow$ | FAD $\downarrow$ | IB-AV $\uparrow$ |
|---|---|---|---|---|
| Landscape | MM-Diffusion | 447 | 5.78 | 0.156 |
| | + MMDisCo | **405** | **5.52** | **0.162** |
| AIST++ | MM-Diffusion | 513 | 2.31 | 0.0897 |
| | + MMDisCo | **450** | **2.17** | **0.0909** |

the parameters of MM-Diffusion. For training, we used the Adam optimizer (Kingma and Ba, 2015) with a learning rate of 1e-3, and the batch size and the number of epochs were set to 16 and 100, respectively. For generation, we used DPM-Solver (Lu et al., 2022) and set its number of function evaluations (NFE) to 20, the standard setting in MM-Diffusion. See Appendix A.3 for more details about the settings.

**Datasets** We conducted experiments on the Landscape (Lee et al., 2022) and AIST++ (Li et al., 2021) datasets. The Landscape dataset contains 928 videos of nine classes of natural scenes, such as fire crackling and waterfall burbling. The AIST++ dataset contains 1020 video clips of street dance with 60 copyright-cleared dancing songs. Note that, in this experiment, we do not use these class labels as input. We trained our discriminator to generate 1.6 second audio-video pairs. Each video comprises ten frames per second at a $64 \times 64$ spatial resolution, and the sampling rate of the audio is 16kHz. We followed the MM-Diffusion setting for audio and video preprocessing and training split.

**Evaluation metrics** We evaluated the generated samples in terms of the cross-modal semantic alignment as well as the fidelity of each modality. To measure the cross-modal semantic alignment, we used the ImageBind score (Girdhar et al., 2023) computed for audio and video embeddings (IB-AV). To evaluate the fidelity of each modality, we used FVD (Unterthiner et al., 2019) for video and FAD (Kilgour et al., 2019) for audio.

**Results** Table 2 shows the quantitative evaluation on both the Landscape and AIST++ datasets. For the fidelity of each single modality, our proposed method improves both FVD and FAD. These results demonstrate that our proposed guidance module captures the training data distribution well and bridges the gap between the distribution of the training data and that of the generated samples. For the alignment score, IB-AV scores are also marginally improved, illustrating our guidance module properly guides generated samples to be well-aligned across modalities.

## 4.3 Experiments with benchmark datasets under the out-of-domain setting

**Setup** We also applied our proposed method to an out-of-domain setting. In this setting, we used a different dataset for training our discriminator compared with the base models. Specifically, we used two pairs of base models, AudioLDM (Liu et al., 2023) / AnimateDiff (Guo et al., 2024) and Auffusion (Xue et al., 2024) / VideoCrafter2 (Chen et al., 2024), to generate audio and video, each pre-trained with respective single-modal large-scale datasets. On top of these independent models, we trained our proposed discriminator using an audio-video dataset to facilitate audio-video alignment of the generated samples from the base models. Note that these base models can accept text input as an additional condition. To enable classifier-free guidance (Ho and Salimans, 2021), we also fed conditional text into our discriminator and trained it with a 10% text dropout rate. We doubled the channel size of each layer of the discriminator to match the larger size of the base models compared to the IND case. We trained our model to generate videos with a spatial size of $256 \times 256$ to match the outputs of the base models. For generation, we adopted classifier-free guidance with its strength set to 2.5 for the AudioLDM, 7.5 for the AnimateDiff, and 8 for the Auffusion and the VideoCrafter2, respectively. See Appendix A.3 for more details.

**Dataset** We conducted experiments with the Landscape (Lee et al., 2022) and VGGSound (Chen et al., 2020) datasets. The VGGSound dataset contains nearly 200K video clips of 300 sound classes.

**Table 3:** Quantitative evaluation under the OOD setting on the Landscape dataset.

| Method | Video | | Audio | | Cross-modal |
|---|---|---|---|---|---|
| | FVD ↓ | IB-TV ↑ | FAD ↓ | IB-TA ↑ | IB-AV ↑ |
| Grand truth | 207 | - | 0.16 | - | 0.247 |
| AnimateDiff → SpecVQGAN | (852) | (0.308) | 10.69 | 0.050 | 0.086 |
| VideoCrafter2 → SpecVQGAN | (700) | (0.309) | 11.63 | 0.049 | 0.074 |
| AnimateDiff → DiffFoley | (852) | (0.308) | 8.58 | 0.053 | 0.113 |
| VideoCrafter2 → DiffFoley | (700) | (0.309) | 9.14 | 0.045 | 0.111 |
| AudioLDM / AnimateDiff | 852 | **0.308** | 8.26 | 0.053 | 0.093 |
| + MMDisCo | **667** | 0.303 | **7.69** | **0.061** | **0.102** |
| Auffusion / VideoCrafter2 | 700 | 0.309 | 8.06 | 0.103 | 0.134 |
| + MMDisCo | **687** | 0.309 | **7.86** | **0.109** | **0.137** |

**Table 4:** Quantitative evaluation under the OOD setting on the VGGSound dataset.

| Method | Video | | Audio | | Cross-modal |
|---|---|---|---|---|---|
| | FVD ↓ | IB-TV ↑ | FAD ↓ | IB-TA ↑ | IB-AV ↑ |
| Grand truth | 256 | - | 1.32 | - | 0.336 |
| AnimateDiff → SpecVQGAN | (739) | (0.295) | 8.65 | 0.064 | 0.084 |
| VideoCrafter2 → SpecVQGAN | (831) | (0.302) | 8.91 | 0.061 | 0.089 |
| AnimateDiff → DiffFoley | (739) | (0.295) | 14.42 | 0.046 | 0.098 |
| VideoCrafter2 → DiffFoley | (831) | (0.302) | 13.11 | 0.060 | 0.105 |
| AudioLDM / AnimateDiff | **739** | **0.295** | 15.5 | 0.107 | 0.121 |
| + MMDisCo | 754 | 0.291 | **12.1** | **0.116** | **0.127** |
| Auffusion / VideoCrafter2 | 831 | 0.302 | 5.32 | 0.190 | 0.192 |
| + MMDisCo | **704** | 0.302 | **4.79** | **0.197** | **0.201** |

Following Yariv et al. (2024), we filtered 60K videos with weak audio-video alignment to enhance the data quality. For both datasets, we resized videos while maintaining the aspect ratio of the spatial resolution and cropped $256 \times 256$ center pixels to create the training dataset. Regarding text captions for the training dataset, we used the original labels for the VGGSound dataset. For the Landscape dataset, we additionally created captions by applying InstructBLIP (Dai et al., 2024) to the first frame of each video because the original labels are too vague to generate plausible samples (see Appendix A.4 for more details).

**Evaluation metrics** We evaluated the generated samples using FAD, FVD, and IB-AV as described in Section 4.2. Additionally, to measure the correspondence between a generated sample and its text condition, we computed the ImageBind score for text-video and text-audio pairs (denoted by IB-TV and IB-TA, respectively).

**Baselines** We compared our method with existing works that can be reproduced and are publicly available. Specifically, we used a sequential approach, Text-to-Video (T2V) generation, followed by Video-to-Audio (V2A) generation, as the baseline for comparison. We used the pre-trained SpecVQGAN (Iashin and Rahtu, 2021) and Diff-Foley (Luo et al., 2023) released in the official repository as a V2A model and generated audios conditioned by the videos from AnimateDiff or VideoCrafter2. For the text conditions of T2V models, we randomly sampled the captions generated by InstructBLIP for the Landscape dataset and the class labels of the VGGSound dataset, which is the same setting as the one for our method.

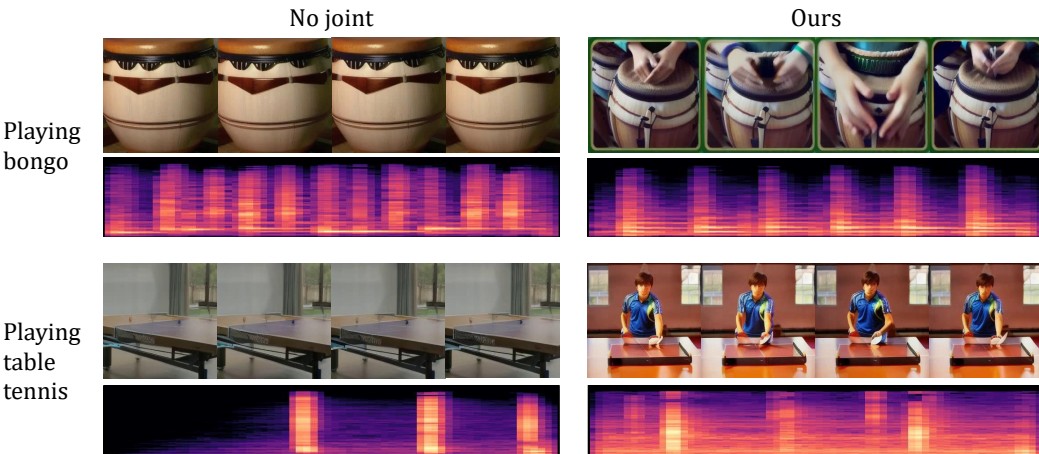

**Figure 3:** Generated samples from AnimateDiff / AudioLDM and ours trained on the VGGSound dataset. We used the captions shown on the leftmost side and the same random seed for both settings to generate samples.

**Results** Tables 3 and 4 show the quantitative evaluation results on the Landscape and VGGSound datasets. For single-modal evaluation, our proposed guidance improves all scores except IB-TV and FVD for the VGGSound. This result indicates our method properly guides the generation process to match the distribution of generated samples with the training dataset, even for the OOD setting. Since IB-TV scores of the base models are substantially higher than IB-TA, we conjecture that the captions are suitable for video generation. Therefore, the base video model can generate videos well-aligned with the text captions, possibly resulting in slight degradation of FVD and IB-TV for the setting of AudioLDM and AnimateDiff but improvement of FAD and IB-TA. For cross-modal metrics, the IB-AV scores are consistently improved, indicating that our method successfully enhances semantic audiovisual alignment. Our method generally delivers superior performance in terms of audio fidelity and IB-AV scores than the T2V followed by V2A generation (T2V → V2A in Tables 3 and 4). We hypothesize that pre-trained T2A models, trained on larger-scale datasets, have the capacity to generate not only high-quality but also sufficiently diverse samples to accommodate various, potentially out-of-domain videos produced by T2V models. Combined with our method, the fidelity of the generated audio and its semantic alignment with the corresponding generated videos are further enhanced. Fig. 3 illustrates the effect of our guidance module qualitatively. Although the generated samples from the base models look reasonable as a single modality, they look unnatural from the perspective of audio-video joint generation because there is no player, but the sound is generated. In contrast, our guidance tends to generate players successfully, and the generated audio is temporally aligned with their motion (see Appendix A.12 for more generated samples).

## 5 CONCLUSION

In this work, we have proposed a novel training-based but model-agnostic guidance module that enables base models to generate well-aligned samples across modalities cooperatively. Specifically, given two pre-trained base diffusion models, we train a lightweight joint guidance module that modifies the scores estimated by the two base models to match the joint data distribution. We show that this guidance can be formulated as the gradient of an optimal discriminator that distinguishes real audio-video pairs from fake pairs generated independently by the base models. We also propose regularizing this gradient using a denoising objective, as in standard diffusion models, which provides a stable gradient of the discriminator. On several benchmark datasets, we empirically show that our proposed method improves alignment scores as well as single-modal fidelity scores without requiring a huge number of parameters compared with the base models.

**Limitation** Since our model is built on top of pre-trained models for each modality, the quality of the generated samples strongly depends on the base models. However, our method benefits from the advancements in each modality's generative modeling. It enables us to integrate new state-of-the-art works into a joint generative model without a huge computational cost.

## 6 ACKNOWLEDGMENT

We sincerely thank Christian Simon, Kazuki Shimada, Michael Yeung, Satoshi Hayakawa, and Shoukang Hu for their helpful feedback. We used computational resources of AI Bridging Cloud Infrastructure (ABCI) provided by the National Institute of Advanced Industrial Science and Technology (AIST) and AIST Solutions.

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

TABLE OF CONTENTS

# A  APPENDIX

## A.1  PROOF OF DISCRIMINATOR GUIDANCE

Our discriminator is trained by Eq. (10). This loss function can be decomposed into the loss function at each timestep as:

$$\mathcal{L}_{\text{disc},t}^{(\theta)} = \mathbb{E}_{q(x_t,y_t)} \left[ \log D_\theta(x_t, y_t, t) \right] + \mathbb{E}_{p_\phi(x_t), p_\psi(y_t)} \left[ \log \left( 1 - D_\theta(x_t, y_t, t) \right) \right] \tag{17}$$

$$= \int_{x_t, y_t} q(x_t, y_t) \log D_\theta(x_t, y_t, t) + p_\phi(x_t) p_\psi(y_t) \log \left( 1 - D_\theta(x_t, y_t, t) \right) \, \mathrm{d}x_t \mathrm{d}y_t. \tag{18}$$

Here, $q(x_t, y_t)$ is a distribution of real paired data, and $p_\phi(x_t) p_\phi(y_t)$ is that of a fake one learned by base models. By Proposition 1 in Goodfellow et al. (2014), for any fixed generators, the optimal discriminator that minimizes Eq. (18) yields:

$$D_{\theta^*}(x_t, y_t, t) \approx \frac{q(x_t, y_t)}{q(x_t, y_t) + p_\phi(x_t) p_\psi(y_t)}. \tag{19}$$

Using this optimal discriminator, we can compute the density ratio between a real and fake data distribution as follows:

$$\frac{q(x_t, y_t)}{p_\phi(x_t) p_\psi(y_t)} \approx \frac{D_{\theta^*}(x_t, y_t, t)}{1 - D_{\theta^*}(x_t, y_t, t)}. \tag{20}$$

Assuming the base models perfectly align with the marginal distribution of the real data (i.e., $q(x_t) = p_\phi(x_t)$ and $q(y_t) = p_\psi(y_t)$), we can compute $\nabla_{x_t} \log q(y_t|x_t)$ and $\nabla_{y_t} \log q(x_t|y_t)$ as follows:

$$\nabla_{x_t} \log q(y_t|x_t) = \nabla_{x_t} \log \frac{q(x_t, y_t)}{q(x_t)} \tag{21}$$

$$= \nabla_{x_t} \log \frac{q(x_t, y_t)}{q(x_t) q(y_t)} \tag{22}$$

$$= \nabla_{x_t} \log \frac{q(x_t, y_t)}{p_\phi(x_t) p_\psi(y_t)} \tag{23}$$

$$\approx \nabla_{x_t} \log \frac{D_{\theta^*}(x_t, y_t, t)}{1 - D_{\theta^*}(x_t, y_t, t)}, \tag{24}$$

$$\nabla_{y_t} \log q(x_t|y_t) = \nabla_{y_t} \log \frac{q(x_t, y_t)}{q(y_t)} \tag{25}$$

$$= \nabla_{y_t} \log \frac{q(x_t, y_t)}{q(x_t) q(y_t)} \tag{26}$$

$$= \nabla_{y_t} \log \frac{q(x_t, y_t)}{p_\phi(x_t) p_\psi(y_t)} \tag{27}$$

$$\approx \nabla_{y_t} \log \frac{D_{\theta^*}(x_t, y_t, t)}{1 - D_{\theta^*}(x_t, y_t, t)}. \tag{28}$$

Therefore, Eqs. (11) and (12) hold.

## A.2  DISCRIMINATOR GUIDANCE FOR THE OOD CASE

In Appendix A.1, we assume $q(x_t) = p_\phi(x_t)$ and $q(y_t) = p_\psi(y_t)$ to derive Eqs. (11) and (12). Here, we prove that they also hold even in the case of $q(x_t) \neq p_\phi(x_t)$ and $q(y_t) \neq p_\psi(y_t)$.

Equations (7) and (8) can be rewritten by using $p_\phi(x_t)$ and $p_\psi(y_t)$ as follows:

$$\nabla_{x_t} \log q(x_t, y_t) = \nabla_{x_t} \log q(x_t) + \nabla_{x_t} \log q(y_t|x_t), \tag{29}$$

$$= \nabla_{x_t} \log p_\phi(x_t) + \nabla_{x_t} \log \frac{q(x_t)}{p_\phi(x_t)} + \nabla_{x_t} \log \frac{q(x_t, y_t)}{q(x_t)} \tag{30}$$

$$= \nabla_{x_t} \log p_\phi(x_t) + \nabla_{x_t} \log \frac{q(x_t, y_t)}{p_\phi(x_t)} \tag{31}$$

$$\approx \nabla_{x_t} \log p_\phi(x_t) + \nabla_{x_t} \log \frac{D_{\theta^*}(x_t, y_t, t)}{1 - D_{\theta^*}(x_t, y_t, t)}, \tag{32}$$

$$\nabla_{y_t} \log q(x_t, y_t) = \nabla_{y_t} \log q(y_t) + \nabla_{y_t} \log q(x_t|y_t) \tag{33}$$

$$= \nabla_{y_t} \log p_\psi(y_t) + \nabla_{y_t} \log \frac{q(y_t)}{p_\psi(y_t)} + \nabla_{y_t} \log \frac{q(x_t, y_t)}{q(y_t)} \tag{34}$$

$$= \nabla_{y_t} \log p_\psi(y_t) + \nabla_{y_t} \log \frac{q(x_t, y_t)}{p_\psi(y_t)} \tag{35}$$

$$\approx \nabla_{y_t} \log p_\psi(y_t) + \nabla_{y_t} \log \frac{D_{\theta^*}(x_t, y_t, t)}{1 - D_{\theta^*}(x_t, y_t, t)}. \tag{36}$$

Therefore, we can use the optimal discriminator as a joint guidance module for the OOD case as long as a discriminator is trained to distinguish real paired samples from fake ones generated by the base models.

### A.3 DETAILS OF EXPERIMENTS

This section provides more details of the settings we used in each experiment.

**Experiments with toy dataset** Table 5 shows the settings for the toy dataset described in Section 4.1. As described in Section 3, residual noise prediction for the denoising step is computed through the gradient of the discriminator with respect to its input, and its dimensionality is two. For evaluation and visualization, we sample 4000 samples.

**Experiments with benchmark datasets** Table 6 shows the settings for real benchmark datasets. We implemented our discriminator based on the official implementation of MM-Diffusion (Ruan et al., 2023). [2] A notable difference is that we only used its encoder part and removed all attention layers from the encoder, which is commonly used in the implementation of diffusion models. Our preliminary experiments found that using attention layers causes unstable discriminator training. Thus, the architecture of our discriminator is a stack of individual 2-stream ResBlocks followed by a linear layer. For the OOD case, we constructed our discriminator on the latent space learned by the base models. To train the discriminator, we sampled 1K samples for the IND settings and 10K samples for the OOD settings in advance.

For evaluation, we generated 2048 samples and computed quantitative metrics. Specifically, we utilized the evaluation code provided by StyleGAN-V (Yu et al., 2022) for FVD,[3] that provided by AudioLDM (Liu et al., 2023) for FAD,[4] and that of IB-score (Girdhar et al., 2023),[5] respectively. During inference, we add the discriminator's gradient to the base models' predictions without scaling. However, when Auffusion and VideoCrafter2 are used as base models, we observed that the gradient norm with respect to the noisy video latents was significantly smaller than the predicted noise from the base models. To address this discrepancy in scale, we apply scaling to the discriminator's gradient with respect to the video latents, but only in this particular setting. We determined the scaling factor by performing a grid search across $[1, 2, 4, 6, 8, 10]$, with a scale of 8 producing the best quantitative results.

---

[2] https://github.com/researchmm/MM-Diffusion
[3] https://github.com/universome/stylegan-v
[4] https://github.com/haoheliu/audioldm_eval
[5] https://github.com/facebookresearch/ImageBind

**Table 5:** Hyperparameters for the base model and discriminator in the experiments with toy dataset (used in Section 4.1).

| Model | Base model | Discriminator (IND) | Discriminator (OOD) |
|---|---|---|---|
| **Hyperparameters** | | | |
| Architecture | MLP | MLP | MLP |
| Input dim | 1 | 2 | 2 |
| # of layers | 5 | 3 | 3 |
| Channel sizes | 16, 64, 256, 64, 16 | 64, 32, 8 | 64, 32, 8 |
| Output dim | 1 | 1 | 1 |
| Normalization | LayerNorm | LayerNorm | LayerNorm |
| Activation | SiLU | SiLU | SiLU |
| Timestep dim | 256 | 64 | 64 |
| Timestep input type | Adaptive | Adaptive | Adaptive |
| Optimizer | Adam | Adam | Adam |
| Learning rate | 0.001 | 0.001 | 0.001 |
| Total batch size | 512 | 512 | 512 |
| Total # of params | 2.2M | 171K | 171K |
| **Diffusion setup** | | | |
| Diffusion steps | 500 | 500 | 500 |
| Noise schedule | linear | linear | linear |
| $\beta_0$ | 0.0001 | 0.0001 | 0.0001 |
| $\beta_T$ | 0.02 | 0.02 | 0.02 |
| **Dataset** | | | |
| $\mu_x$ of GMM | [-3, -1.5, 0, 1.5, 3] | [-3, -1.5, 0, 1.5, 3] | [2.25, -2.25, 2.25, -2.25, 0] |
| $\mu_y$ of GMM | - | [1.5, -3, 3, 0, -1.5] | [2.25, -2.25, 2.25, -2.25, 0] |
| $\sigma$ of GMM | 0.1 | 0.1 | 0.1 |
| # of fake samples | - | 500 | 500 |
| # of real samples | 500 | 500 | 500 |

We utilized 16GB Nvidia V100 × 4 GPUs for the IND case and 40GB Nvidia A100 × 8 GPUs for the OOD case. The training time for the IND case was about 1.5 hours with both Landscape and AIST++ datasets, and that for the OOD case was about two hours with the Landscape dataset and 30 hours with the VGGSound dataset. All evaluation was performed by 16GB Nvidia V100 × 4 GPUs. It takes about one hour for the IND case and about five hours for the OOD case.

### A.4 CAPTIONING PROCESS OF LANDSCAPE DATASET FOR THE OOD SETTING

We found that the original labels added to the Landscape dataset are too ambiguous for the base models to generate reasonable samples. For instance, the generated results with the caption "wind noise" tend to look like just a noise video and audio, and evaluation for these samples is unstable and not desired. Therefore, we obtained simple captions for each video using InstructBLIP (Dai et al., 2024) and used them only for the OOD experiment with the Landscape dataset. Specifically, we extracted the first frame of each video and passed it to the InstructBLIP. As an instruction for the captioning, we used the sentence: "*Write a short sentence for the image starting with 'The image captures a scene with'.*" and generated up to 128 characters for each video. Note that these captions are generated only from videos without audio. They tend to describe only visual content and provide less information about audio. Therefore, these captions may not be suitable for audio generation by AudioLDM, as we show in Tables 3 and 4 where IB-TA is worse than IB-TV. These results show that the text condition strongly affects the quality of generated results. Designing and manipulating the input condition is also important for the joint generation, which we leave for future work.

### A.5 VISUALIZATION OF THE EFFECT BY EACH LOSS FUNCTION WITH TOY DATASETS

Figure 4 visualizes the effect of each loss function. The top row shows the IND case, and the bottom shows the OOD case. At each row, the leftmost column shows the samples from the target distribution,

**Table 6:** Hyperparameters used in the experiments with benchmark datasets (used in Sections 4.2 and 4.3). †
We jointly generate audio and video by MM-Diffusion in the IND case.

| Model | Discriminator (IND) | Discriminator (OOD) |
|---|---|---|
| **Hyperparameters** | | |
| Audio base model | MM-Diffusion[†] | AudioLDM / Auffusion |
| Video base model | MM-Diffusion[†] | AnimateDiff / VideoCrafter2 |
| Architecture | ResBlocks | ResBlocks |
| Audio input dims | 1, 25600 | 8, 50, 16 |
| Video input dims | 3, 16, 64, 64 | 4, 16, 32, 32 |
| Video fps | 10 | 8 |
| Audio fps | 16k | 16k |
| Duration (sec) | 1.6 | 2.0 |
| # of ResBlocks per resolution | 2 | 4 |
| Audio conv type | 1d | 2d |
| Audio conv dilation size | $1, 2, 4, ..., 2^{10}$ | 1 |
| Video conv type | 2d + 1d | 2d + 1d |
| Channels | 128 | 256 |
| Channel multiplier | 1, 2, 4 | 1, 2, 4 |
| Audio downsample factor | 4 | (1, 2) |
| Video downsample factor | (1, 2, 2) | (1, 2, 2) |
| Output dim | 1 | 1 |
| Normalization | GroupNorm | GroupNorm |
| Activation | SiLU | SiLU |
| Timestep dim | 128 | 256 |
| Timestep input type | Adaptive | Adaptive |
| Conditional input | - | Text |
| Condition drop rate | - | 0.1 |
| Text embed dim for video | - | 768 |
| Text embed dim for audio | - | 512 |
| Text embed input type | - | Add |
| Optimizer | Adam | Adam |
| Learning rate | 0.001 | 0.001 |
| Total batch size | 16 | 32 |
| Total training epochs | 100 | Landscape: 100 / VGGSound: 10 |
| Total # of params for base models | 133M | 2.15B / 2.86B |
| Total # of params for discriminator | 13M | 132M |
| **Diffusion setup** | | |
| Diffusion steps | 1000 | 1000 |
| Noise schedule | linear | scaled linear |
| $\beta_0$ and $\beta_T$ for audio | 0.0001 / 0.02 | 0.0015 / 0.0195 |
| $\beta_0$ and $\beta_T$ for video | 0.0001 / 0.02 | 0.00085 / 0.012 |
| **Inference parameters** | | |
| Sampler | DPM-Solver | DPM-Solver ++ |
| NFE | 20 | 50 |
| Order | 3 | 2 |
| Audio text guidance scale | - | 2.5 / 8.0 |
| Video text guidance scale | - | 7.5 / 8.0 |
| Audio joint guidance scale | 1.0 | 1.0 / 1.0 |
| Video joint guidance scale | 1.0 | 1.0 / 8.0 |

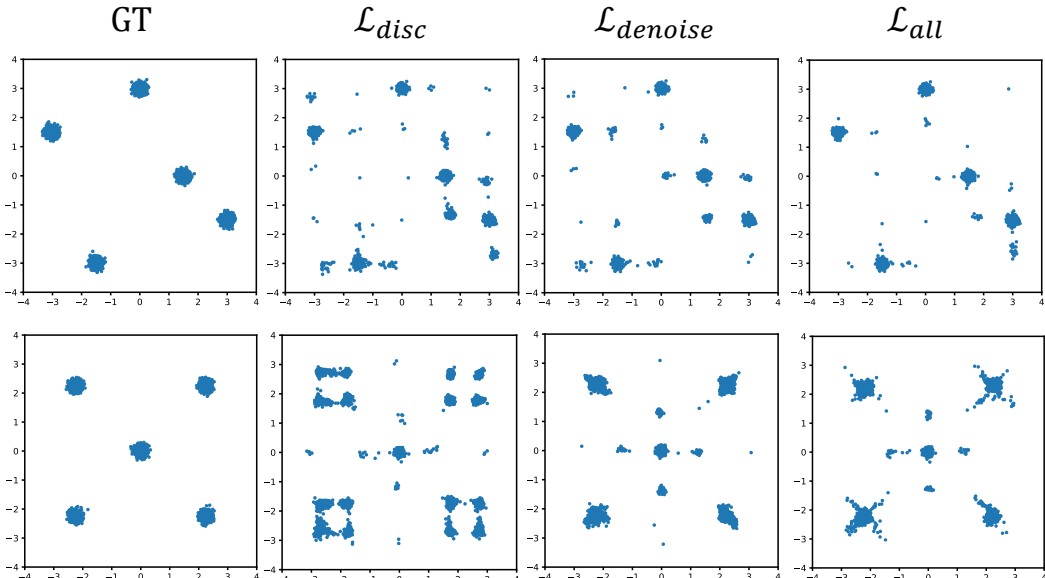

**Figure 4:** Visualization of the generated samples across loss functions used to train our guidance module with the toy dataset. The top row shows the IND setting, and the bottom shows the OOD setting.

**Table 7:** Ablation study about the loss functions on the landscape dataset.

| Settings | Video | | Audio | | Cross-modal |
|---|---|---|---|---|---|
| | FVD ↓ | IB-TV ↑ | FAD ↓ | IB-TA ↑ | IB-AV ↑ |
| Dataset | 207 | - | 0.16 | - | 0.247 |
| No Joint | 852 | 0.308 | 8.26 | 0.053 | 0.093 |
| $\mathcal{L}_{\text{disc}}$ | 818 | **0.305** | 7.80 | 0.060 | **0.102** |
| $\mathcal{L}_{\text{denoise}}$ | 700 | 0.303 | **7.19** | **0.064** | 0.088 |
| $\mathcal{L}_{\text{all}}$ | **667** | 0.303 | 7.69 | 0.061 | **0.102** |

and the others show generated samples with our guidance modules trained by different loss functions. Note that the NNL at each setting has been shown in Table 1. In the IND case, both $\mathcal{L}_{\text{disc}}$ and $\mathcal{L}_{\text{denoise}}$ work similarly, whereas in the OOD case, the guidance trained by only $\mathcal{L}_{\text{disc}}$ can roughly concentrate the samples at four corners but struggles to force them be in a single peak. In contrast, combining with $\mathcal{L}_{\text{denoise}}$ drastically improves this issue, and using both of them yields the best result.

## A.6 ABLATION STUDY ABOUT THE LOSS FUNCTIONS ON REAL DATA

We conducted an ablation study about the loss functions on the landscape dataset. Table 7 shows quantitative results of our model trained with $\mathcal{L}_{\text{disc}}$, $\mathcal{L}_{\text{denoise}}$, and $\mathcal{L}_{\text{all}}$. We used AudioLDM / AnimateDiff as base models in this experiment. The model trained on $\mathcal{L}_{\text{disc}}$ achieves better IB-AV and shows good performance in terms of the cross-modal semantic alignment, while it shows moderate improvements in the fidelity scores. On the other hand, the one trained on $\mathcal{L}_{\text{denoise}}$ achieves better fidelity scores while suffering from lower cross-modal alignment scores. Compared to these, the one trained on $\mathcal{L}_{\text{all}}$ achieves both better fidelity scores and cross-modal alignment, which has the best of both worlds.

## A.7 VALIDATION OF THE PARAMETER SIZE AND THE TRAINING EPOCH

We explored the effect of the parameter size of the discriminator as well as the number of training epochs in the IND setting with the Landscape dataset.

**Table 8:** Ablation study for the channel size $C$ of the discriminator with the number of ResBlocks per resolution $L$ fixed to 2.

| Architecture | | Metrics | | |
|---|---|---|---|---|
| Settings | # of params | FVD ↓ | FAD ↓ | IB-AV ↑ |
| Base model (MM-Diffusion) | 133M | 447 | 5.78 | 0.156 |
| $C = 32, L = 2$ | 798K | 423 | 5.60 | 0.159 |
| $C = 64, L = 2$ | 3.2M | 410 | 5.48 | 0.161 |
| $C = 128, L = 2$ | 12.7M | 405 | 5.52 | **0.162** |
| $C = 256, L = 2$ | 50.7M | **399** | **5.46** | 0.160 |

**Table 9:** Ablation study for the number of ResBlocks per resolution $L$ of the discriminator with the channel size $C$ fixed to 32 and 128.

| Architecture | | Metrics | | |
|---|---|---|---|---|
| Settings | # of params | FVD ↓ | FAD ↓ | IB-AV ↑ |
| Base model (MM-Diffusion) | 133M | 447 | 5.78 | 0.156 |
| $C = 32, L = 1$ | 401K | **421** | 5.65 | 0.159 |
| $C = 32, L = 2$ | 798K | 423 | 5.60 | 0.159 |
| $C = 32, L = 4$ | 1.6M | 425 | **5.51** | **0.160** |
| $C = 128, L = 1$ | 19.0M | 415 | 5.60 | 0.161 |
| $C = 128, L = 2$ | 12.7M | **405** | 5.52 | **0.162** |
| $C = 128, L = 4$ | 25.3M | 407 | **5.43** | **0.162** |

Table 8 shows the effect of increasing the channel size (denoted by $C$) with the number of ResBlocks per resolution fixed. As we increase the channel size, performance substantially increases for both the single-modal fidelity score and the multimodal alignment score. We observed that the multimodal alignment is slightly degraded at $C = 256$ and concluded that the setting with $C = 128$ is the most appropriate for multimodal guidance, requiring only less than 10% additional parameters. Note that our proposed method successfully improves the base model's performance even when we only used less than 1% additional parameters (the setting with $C = 32$).

Table 9 shows the effect of increasing the number of ResBlocks per resolution (denoted by $L$) with the channel size fixed. We did experiments on the settings with $C = 32$ and $C = 128$. For both settings, increasing $L$ improves FAD mainly, while the differences of other metrics are marginal. We suppose this may be the effect of the increase in the receptive field. Since MM-diffusion is trained on the raw data space (pixel for video and waveform for audio), the size of the time axis for audio is extremely large. Therefore, following the implementation of MM-diffusion, we used dilated convolutions for the audio branch to capture features at multiple scales. Using more ResBlocks containing dilated convolutions may improve the audio encoder's capacity, resulting in improved audio performance.

Figure 5 shows the performance of our guidance module over the number of training epochs. We used $C = 128$ and $L = 2$ for the discriminator and trained it longer. For evaluation, we generated samples with five different random seeds and computed the average and standard deviation of each metric. Our guidance module converged around 100 epochs, and we did not observe clear improvement by training further.

## A.8 DISCRIMINATOR STRUCTURE VARIANTS

Our discriminator basically consists of two modules in terms of functionality: i) feature extraction module and ii) feature fusion module. We used a stack of ResBlock layers to extract modality-specific features for i) and channel-wise concatenation of pooled features along the time axis, followed by an MLP to handle crossmodal interaction for ii). We denote these default structures as "Res" for i) and "MLP" for ii). To further improve the performance of joint generation, we tested integrating transformer architecture into our discriminator, inspired by its recent success (Vaswani,

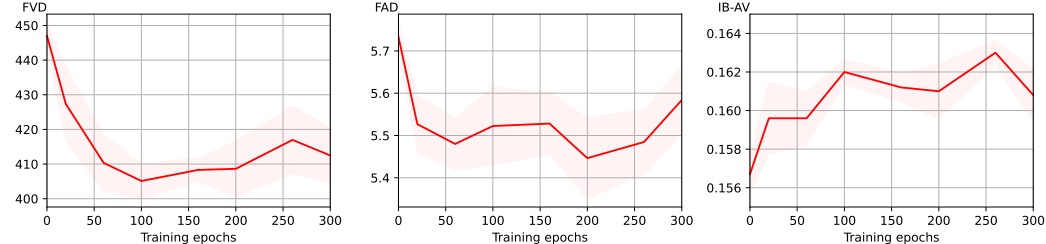

**Figure 5:** Ablation study for the training epochs. We trained the discriminator with $C = 128$ and $L = 2$ for 300 training epochs. We generated samples with five different random seeds and computed their average and std.

**Table 10:** Ablation study for the different discriminator designs. For the notation of the architectures, we denote "i) / ii)" where i) is the architecture of the feature extractor module and ii) is the feature fusion module described in Appendix A.8. We used "Auffusion / VideoCrafter2" as the base model.

| Discriminator architecture | Metrics | | | | |
| --- | --- | --- | --- | --- | --- |
| | FVD ↓ | IB-TV ↑ | FAD ↓ | IB-TA ↑ | IB-AV ↑ |
| No discriminator | 831 | 0.302 | 5.32 | 0.190 | 0.192 |
| Res / MLP (default) | **704** | 0.302 | 4.79 | 0.197 | 0.201 |
| Res / Naive transformer | 709 | **0.303** | **4.69** | 0.200 | 0.202 |
| Res / Frieren | 710 | 0.302 | 4.77 | **0.202** | **0.205** |
| ViT / MLP | 730 | 0.301 | 5.07 | 0.192 | 0.199 |
| ViT / Naive transformer | 716 | 0.302 | 5.09 | 0.194 | 0.202 |
| ViT / Frieren | 720 | 0.302 | 5.06 | 0.194 | 0.202 |

2017; Dosovitskiy et al., 2021; Peebles and Xie, 2023; Wang et al., 2024). Specifically, we used two input types for the transformer encoder to replace the module ii). The audio and video features extracted from i) have shapes $(B, F_a, C)$ and $(B, F_v, C)$, respectively, where $B$ is the batch size, $F_a$ and $F_v$ are the numbers of frames in audio and video respectively, and $C$ is the channel size. For the first design, we concatenated these two features along the frame axis to get a crossmodal feature of the shape $(B, F_a + F_v, C)$ (denoted as "Naive Transformer"). For the second design, following the design proposed by Wang et al. (2024), we upsampled shorter features and concatenated these two features along the channel axis to get a crossmodal feature of the shape $(B, \max(F_a, F_v), 2C)$ (denoted as "Frieren"). For both settings, the input is supplemented with a learnable positional embedding and a special token, which is used for the final output, and the transformer encoder with four layers is applied. We implemented the transformer using xFormers[6] and trained both discriminator designs on top of Auffusion / VideoCrafter2 as base models for ten epochs on the VGGSound dataset. Table 10 shows the results. Overall, both designs improve the fidelity and crossmodal alignment. This result indicates that a more sophisticated discriminator structure leads to better performance of crossmodal generation. Although we also briefly tested replacing ResBlock layers with vision transformer architecture (Dosovitskiy et al., 2021; Arnab et al., 2021) (the 4D audio latents are processed by ViT, and the 5D video latents are processed by ViViT) for i), we do not observe performance improvement (denoted as "ViT" in Table 10). Further exploration of the discriminator structures would be an interesting direction for future work.

## A.9 COMPUTATIONAL COST FOR THE TRAINING AND INFERENCE

To highlight the computational efficiency of our proposed method, we compared the computation time for the training and inference of base models, both with and without our method. For this experiment, we used the same network configurations as described in Sections 4.2 and 4.3, with a batch size of 8, and conducted all runs on a single Nvidia H100 GPU.

---

[6]https://github.com/facebookresearch/xformers

**Table 11:** Computational cost for the training. The Computation time of forward and backward execution for the base models and our discriminator are shown. Those of the base models include the computation time required for both audio and video. All numbers except "Speed-up Ratio" show the computation time in seconds.

| | Base Models | | |
|---|---|---|---|
| | MM-Diffusion | AudioLDM / AnimateDiff | Auffusion / VideoCrafter2 |
| Base Fwd | 0.273 (0.009) | 0.254 (0.057) | 0.349 (0.020) |
| Base Bwd | 0.660 (0.001) | 0.505 (0.007) | 0.952 (0.013) |
| Disc Fwd | 0.054 (0.006) | 0.081 (0.007) | 0.081 (0.006) |
| Disc Bwd | 0.135 (0.009) | 0.186 (0.005) | 0.186 (0.007) |
| Baseline 1step | 0.933 (0.009) | 0.759 (0.057) | 1.301 (0.023) |
| Ours 1step | **0.462** (0.014) | **0.521** (0.058) | **0.616** (0.022) |
| Speed-up Ratio | 202% | 146% | 211% |

**Table 12:** Computation time for the inference with and without the proposed method. All numbers except "Overhead" show the time in seconds for generating one pair of audio and video ([second / sample]).

| | Base Models | | |
|---|---|---|---|
| | MM-Diffusion | AudioLDM / AnimateDiff | Auffusion / VideoCrafter2 |
| w/o ours | 1.09 (0.03) | 3.10 (0.02) | 4.45 (0.03) |
| w/ ours | 1.25 (0.04) | 3.77 (0.09) | 4.91 (0.04) |
| Overhead | 15% | 22% | 10% |

Table 11 shows the elapsed time for each base model's forward and backward computations and our discriminator's during a single training step. Each configuration was run 500 times, with the mean and standard deviation for each value reported in the table. As a baseline, we measured the computation time for the single training step of all base models (represented as the sum of Base Fwd and Base Bwd in Table 11). Note that this baseline reflects the cost for fine-tuning base models without incorporating additional learnable layers (Hu et al., 2022) or cross-modal modules (Tang et al., 2023), thereby serving as a lower-bound estimate for the training costs associated with building a joint generation model by simply combining base models. Our method accelerates training by approximately 1.5 to 2 times compared to the baselines. These results demonstrate that our method integrates base models into a joint model in a computationally efficient manner.

Table 12 shows the time required to sample a pair of audio and video. Due to the additional computation required by the discriminator, the inference time is slightly longer (around 10 to 20 %) compared to those of the base models. This overhead can be reduced by designing a more computationally efficient network architecture.

### A.10 COMPARISON WITH BASELINES ON THE SEMANTIC AND TEMPORAL CROSS-MODAL ALIGNMENT

In this section, we show the additional quantitative comparison between our method and baselines in terms of the cross-modal alignment. In addition to the semantic alignment evaluation using the ImageBind score, we computed the AV-Align score (Yariv et al., 2024) for the temporal alignment evaluation. First, we briefly explain the AV-Align score and some adjustments to achieve stable computation in our experimental settings in Appendix A.10.1. Then, we show the results in Appendix A.10.2.

#### A.10.1 COMPUTATION OF THE AV-ALIGN SCORE

The AV-Align score (Yariv et al., 2024) is defined as Intersection-over-Union (IoU) between the detected audio onsets and the peaks of the optical flow in the video. Let $\mathcal{A}$ be a set of the onset peaks detected from the audio, and $\mathcal{V}$ be a set of the peaks of the optical flow in the video.

**Table 13:** Comparison of the cross-modal alignment scores under the OOD setting on the VGGSound dataset.

| Method | AV-Align ↑ | IB-AV ↑ |
|---|---|---|
| Grand truth | 0.296 | 0.336 |
| AnimateDiff → SpecVQGAN | 0.288 | 0.084 |
| VideoCrafter2 → SpecVQGAN | 0.291 | 0.089 |
| AnimateDiff → DiffFoley | 0.295 | 0.098 |
| VideoCrafter2 → DiffFoley | 0.319 | 0.105 |
| AudioLDM / AnimateDiff | 0.320 | 0.121 |
| + MMDisCo (Res / MLP) | **0.330** | 0.127 |
| Auffusion / VideoCrafter2 | 0.268 | 0.192 |
| + MMDisCo (Res / MLP) | 0.287 | 0.201 |
| + MMDisCo (Res / Frieren) | 0.292 | **0.205** |

$$\text{AV-Align} = \frac{1}{2|\mathcal{A} \cup \mathcal{V}|} \left( \sum_{a \in \mathcal{A}} \mathbf{1}[a \in \mathcal{V}] + \sum_{v \in \mathcal{V}} \mathbf{1}[v \in \mathcal{A}] \right), \tag{37}$$

where $\mathbf{1}[a \in \mathcal{V}]$ and $\mathbf{1}[v \in \mathcal{A}]$ are the number of the valid peaks of audio and video. The peak is considered valid if placed within three frames in the other modality.

In the official implementation, the AV-Align score is computed by the following equation:

$$\text{AV-Align} = \frac{|A \cap V|}{|A| + |V| - |A \cap V|}, \tag{38}$$

where they use the value of $\sum_{a \in \mathcal{A}} \mathbf{1}[a \in \mathcal{V}]$ for the intersection $|A \cap V|$. By its definition, the value of the IOU should be in the range of $[0, 1]$. If the audio peaks do not have one-to-many correspondences with video peaks, the computed value with $|A \cap V| \leftarrow \sum_{a \in \mathcal{A}} \mathbf{1}[a \in \mathcal{V}]$ is always in $[0, 1]$.

To ensure no one-to-many correspondence occurs, we need to set the peak detection interval properly. If the interval of the peaks of either modality is sufficiently smaller than the other, one-to-many correspondences are likely to occur. In the configuration of Yariv et al. (2024), they used 24 fps videos (the detection interval is about 42 ms) and set the onset detection interval as 32 ms. On the other hand, we used eight fps videos for the evaluation, which are three times larger intervals than the official configuration. When we compute the AV-align score with its configuration, the computed value occasionally exceeds 1. To compensate for this gap, we set the onset detection interval as 96 ms. We did not observe the value exceeding 1 in our experiment with this configuration.

### A.10.2 EVALUATION ON THE TEMPORAL AND SEMANTIC ALIGNMENT

We computed AV-align scores for the base models and those with our proposed method on the VGGSound dataset. All configurations are the same as Section 4.3.

Table 13 shows the AV-Align and the IB-AV scores of our method and base models. For the AV-Align, "AudioLDM / AnimateDiff + ours (Res / MLP)" achieves the best score among all models. While it also achieves a better IB-AV score than existing models, the IB-AV score significantly lags behind GT's. Using more sophisticated base models ("Auffusion / VideoCrafter2" + ours) significantly improves the IB-AV score with a comparable AV-Align score to the GT. Notably, our proposed method successfully improves both AV-Align and IB-AV scores of both base models.

We also observed that using a more sophisticated network structure ("VideoCrafter2 / Auffusion + ours (Res / Frieren)") further improves both scores. How to design the discriminator to improve the specific types of alignment would be a possible direction of future work. Additionally, how to create a dataset would be a possible direction to improve the alignment. For example, Luo et al. (2023)

**Table 14:** Subjective evaluation comparing the generated results of base models (Auffusion / VideoCrafter2) with those of our method. The evaluation considered four aspects: Video Quality (VQ), Audio Quality (AQ), Semantic Alignment between audio and video (SA), and Temporal Alignment between audio and video (TA). "N/A" indicates the percentage of evaluators who were unable to determine which sample was superior.

| Method | VQ | AQ | SA | TA |
|---|---|---|---|---|
| Auffusion / VideoCrafter2 | 4.2% | 12.5% | 6.9% | 9.7% |
| + MMDisCo | **94.4%** | **44.4%** | **81.9%** | **80.6%** |
| N/A | 1.4% | 43.1% | 11.1% | 9.7% |

proposes using time-shifted audio and video pairs to train audio-visual latent space, which is used for the condition of their generative model, to improve the temporal alignment. In our experiments, we only used generated samples from base models for the fake samples. Adding such fake pairs as simulated temporal misalignment for the fake samples would improve the alignment of generated samples, which we leave for future work.

## A.11 SUBJECTIVE EVALUATION

In addition to the objective evaluation, we conducted a user study to assess the perceptual quality. We generated samples using Auffusion / VideoCrafter2, both with and without our method, employing 10 labels randomly sampled from the VGGSound dataset. This resulted in 10 pairs of two videos, where one was produced by the base models and the other by the base models with our method. We asked human evaluators to answer the following four questions per each pair of videos:

1. Which video is of higher quality regardless of the audio?
2. Which audio is of higher quality regardless of the video?
3. Which pair of audio and video has better semantic alignment?
4. Which pair of audio and video has better temporal alignment?

For each question, evaluators were given three response options: "Video A is better," "Video B is better," and "N/A." We collected 320 answers in total from 8 evaluators. Table 14 provides a summary of the results. Our method consistently enhances the quality of generated samples across all aspects, which aligns with the objective evaluation results presented in Section 4 and Appendix A.10.2.

## A.12 GENERATED AUDIO AND VIDEO PAIRS

This section shows the results generated from the base models and our method. Note that, to evaluate the effectiveness of our proposed method visually, we used the same random seed for the base models and our method, with which we can expect them to provide similar samples.

Figure 6 shows the results of unconditional generation from MM-Diffusion and that with our guidance module. Since MM-Diffusion was already trained by the same dataset (i.e., the IND setting), the generated pairs of audio and video look well-aligned in their semantics. However, our guidance module improves the quality of detail at each modality or forces the generated audio and video to be temporally aligned, which is coherent with the quantitative evaluation results.

Figure 7 shows the results of text-conditional generation from base models (AudioLDM / Animate-Diff) and our method. In this case, since the base models independently generate audio and video, the generated audio and video are not always well-aligned (i.e., the OOD setting). Our guidance module substantially improves the quality of generated samples from base models compared to the IND setting. As described in Section 4.3, our guidance tends to generate players successfully, and the generated audios are also temporally aligned with their motion.

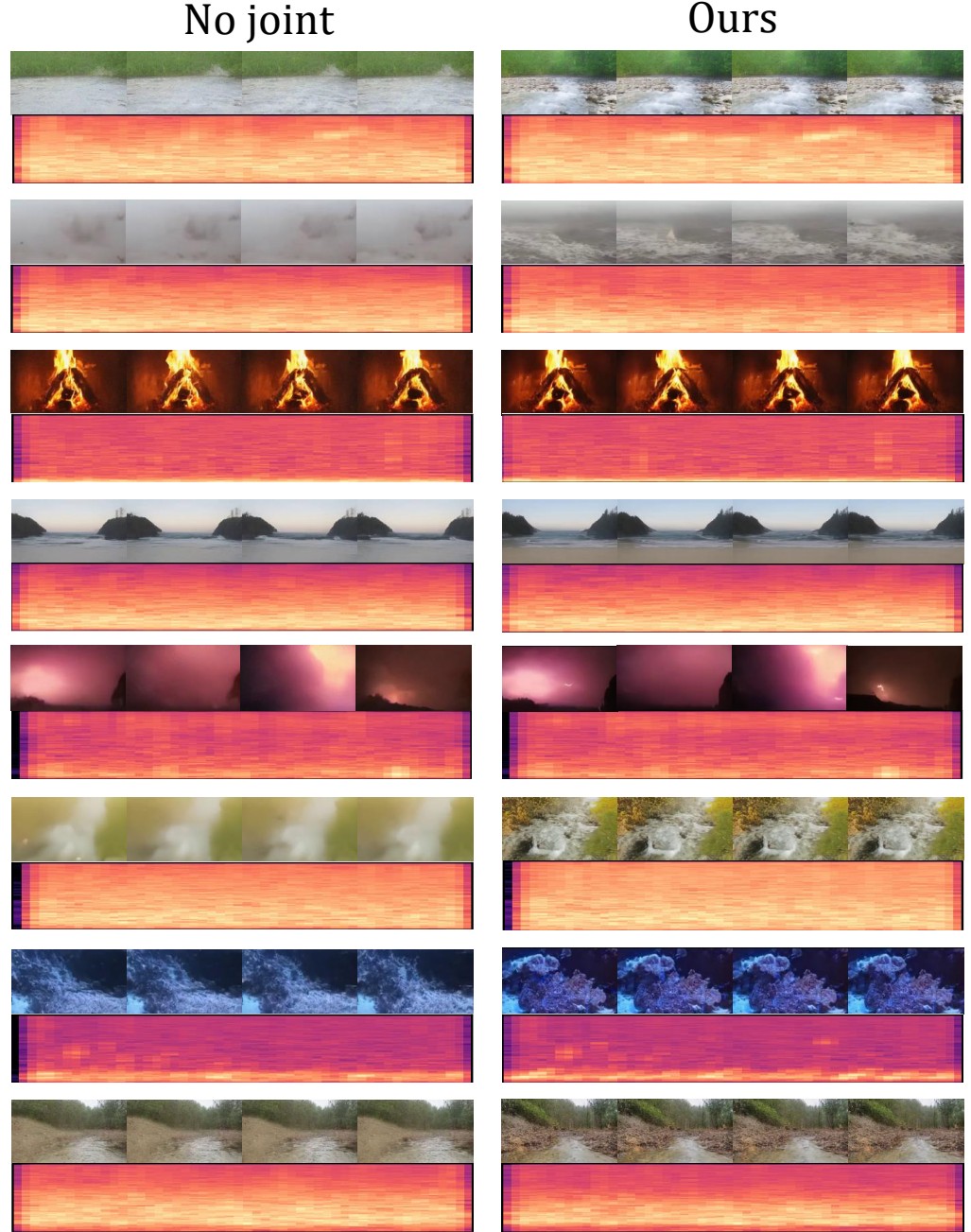

**Figure 6:** More generated samples from MM-Diffusion and ours trained on the Landscape dataset in the IND setting. We used the same random seed for both settings, and the generated results at each row should be similar.

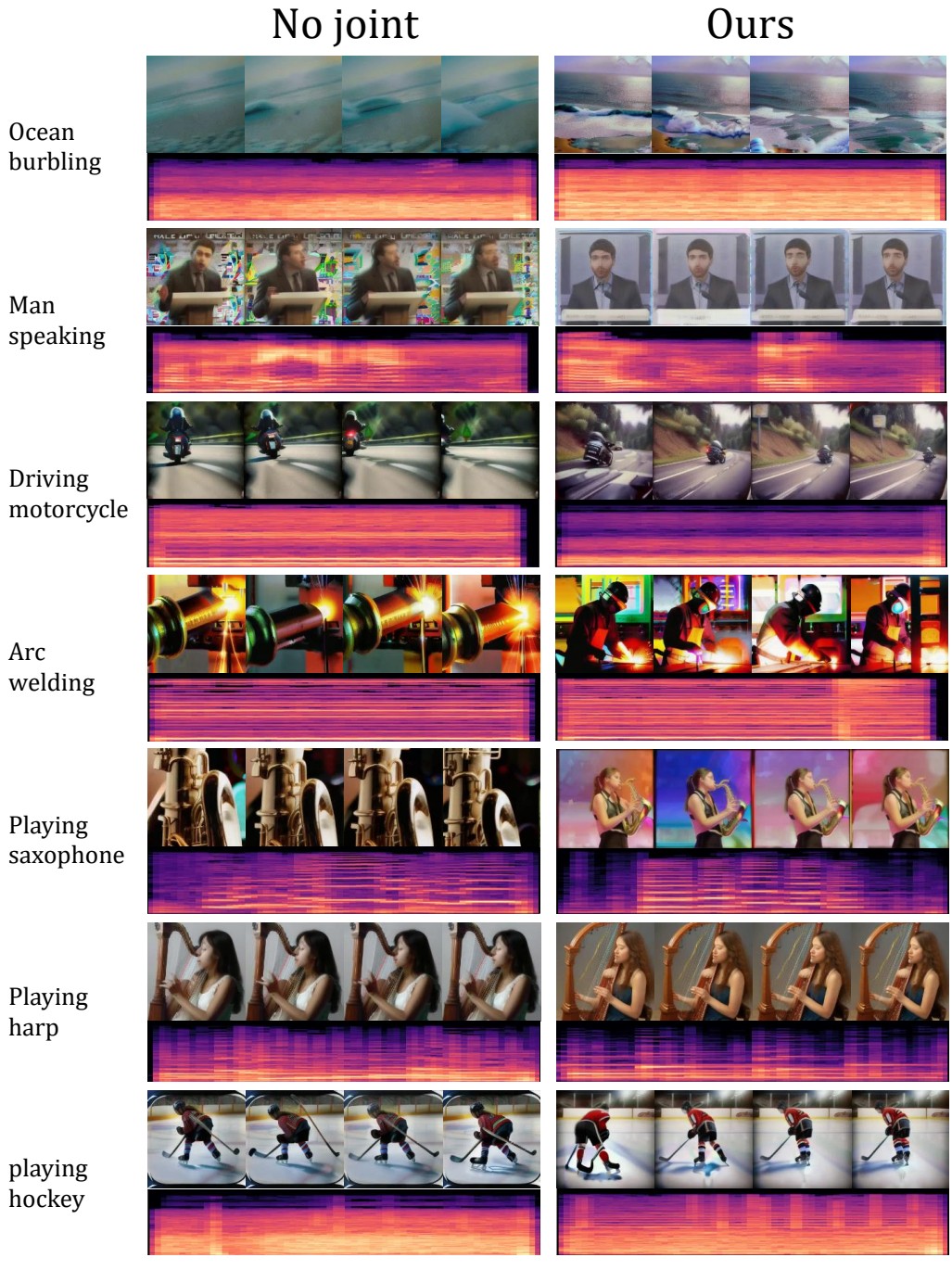

**Figure 7:** More generated samples from AnimateDiff / AudioLDM and our method trained on the VGGSound dataset in the OOD setting. The captions given for the generation of each sample are shown in the leftmost column, and we used the same random seed for both settings.

