# OpenReview forum: "MMDisCo: Multi-Modal Discriminator-Guided Cooperative Diffusion for Joint Audio and Video Generation"
_ICLR.cc/2025/Conference — ICLR 2025 Poster_

### Official Review · Reviewer_U5ht · 2024-11-01

**Soundness:** 3
**Presentation:** 2
**Contribution:** 2
**Rating:** 6
**Confidence:** 3

**Summary:**

This paper presents a novel method for creating an audio-video generative model using pre-trained generative models for audio and video separately, and then guiding these models to work together using a discriminator.

The key innovation is a lightweight joint guidance module (an audio-video discriminator), which adjusts the scores estimated by the base models to match the score of the joint distribution over audio and video. This guidance is computed using the gradient of the discriminator, which distinguishes between real and fake audio-video pairs. The authors also introduce a regulation loss function that stabilizes the gradient of the discriminator.

**Strengths:**

1. The introduced joint module is lightweight and can be adapted to different single-modal generative models.

2. Authors conducted thorough experiments, including toy datasets, and in-domain and out-domain settings, demonstrating the effectiveness of the proposed method.

**Weaknesses:**

1. Statement of datasets and evaluation metrics are duplicated in Sec 4.2 and 4.3.

2. In the section of related work, the author referred to Xing et al. (2024) as their similar work. However, neither quantitative nor qualitative comparisons are conducted with this work.

**Questions:**

1. The training of generative adversarial models is always easy to explode. Did the authors face similar problems when optimizing the discriminator?

2. Since the additional training module is lightweight, I think the training procedure can be efficient. Can the author provide detailed training consumption required for training the discriminator? How much inference time does it increase?

---

> ### Author Response · Authors · 2024-11-20
> **Rebuttal Response (1/2)**
>
> We appreciate the constructive feedback from the reviewer. We would like to answer the questions and the comments below.
>
> ## Duplicated statement about evaluation metrics and datasets in Sec 4.2 and 4.3 (Weakness 1)
> Thank you for your suggestion.
> Based on your comment, we carefully checked our manuscript but could not find any duplication in the evaluation metrics and datasets described in Sections 4.2 and 4.3.
> Section 4.2 explains the details of the Landscape dataset, FVD, FAD, and IB-AV scores, which are used in both Sections 4.2 and 4.3.
> Then, Section 4.3 explains the details of the VGGSound dataset, IB-TV and IB-TA scores, which are used only in Section 4.3.
>
> ## Comparison between "Seeing and Hearing" (Weakness 2)
> Since they do not release the code for joint video and audio generation in their official repository [1], we cannot directly compare their method and ours.
> Therefore, here, we compare our method with Seeing and Hearing (SaH) from the perspective of theoretical difference and clarify the benefit of our method by using the reported numbers in their paper.
>
> Our method differs from SaH by explicitly guiding base models' generation process toward a specific joint distribution.
> Specifically, we formulate our method to bridge the gap between an independent distribution $p(x)p(y)$ and a joint distribution $p(x, y)$.
> We assume that the training data is a sample of the target distribution $p(x, y)$ and derive the loss function to match the distribution learned by the base models with the target one.
> Therefore, the generation with our proposed method can be considered sampling from the target distribution $p(x, y)$.
> In contrast, SaH guides the generated samples to be closed in the latent space of a pre-trained foundation model, ImageBind.
> Although they successfully improve the audio and video alignment, the distribution of its generated samples is not explicitly defined.
> This difference makes our method preferable regarding two aspects: i) effectiveness as a guidance module for a joint generation model and ii) customization of base models for the specific dataset.
>
> From the perspective of i), our guidance can be applied to the joint generation model to boost its performance.
> As shown in Table 2, our method successfully improves both FAD and FVD of MM-Diffusion.
> In contrast, SaH struggles to improve both metrics of MM-Diffusion (FAD improves from 7.7 to 6.4 while FVD degrades from 1141 to 1174 as they reported in Table 1 in their paper).
> Our method provides a similar benefit to the classifier or classifier-free guidance commonly used in conditional generation.
>
> From the perspective of ii), our method successfully guides the same base models toward different joint distributions using several datasets (e.g., Landscape and VGGSound datasets).
> Tables 3 and 4 demonstrate this success as our method improves the FAD and FVD computed on each target dataset.
> Note that it is difficult to achieve this improvement by SaH because there is no guarantee that the distribution of generated samples by their method is closed to the target distribution.
> Our method allows us to customize base models for different datasets to achieve sampling from the specific target distribution.
>
> [1] https://github.com/yzxing87/Seeing-and-Hearing?tab=readme-ov-file#todo

---

> > ### Author Response · Authors · 2024-11-20
> > **Rebuttal Response (2/2)**
> >
> > ## Training stability of the proposed discriminator (Question 1)
> > Thank you for the question.
> > We did not observe any instability in training our discriminator.
> > This is because our proposed method only trains a discriminator to distinguish fake video-audio pairs generated by fixed base models from real ones of the dataset (i.e., our method does not train a generator).
> > In contrast, the training of typical GANs requires unstable min-max optimization.
> >
> >
> > ## Training consumption required for training discriminator (Question 2)
> > Thank you for the question.
> >
> > > Can the author provide detailed training consumption required for training the discriminator?
> >
> > Yes. In short, our proposed method can be trained around 1.5 to 2 times faster than baselines.
> > We measured the computation time for the forward and backward of the base models and our discriminator, and the times for fine-tuning base models without any extension and training of our method were compared.
> > The summarized results are shown below.
> >
> > || MM-diffusion | AudioLDM / AnimateDiff | Auffusion / VideoCrafter2 |
> > | :---: | :---: | :---: | :---: |
> > | Baseline training 1step [sec] | 0.933 | 0.759 | 1.301 |
> > | Ours training 1step [sec] | **0.462** | **0.521** | **0.616**|
> > | Speed-up Ratio | 202% | 146% | 211% |
> >
> > Note that we include forward passes through the frozen modules in "Ours training 1step".
> > These results concretely demonstrate the computational efficiency of our proposed method.
> > Please see the updated manuscript for more details.
> > The complete numbers for each part of the computation are shown in Table 11, and a detailed discussion is provided in appendix A.9.
> >
> > > How much inference time does it increase?
> >
> > We also measured the inference overhead induced by our proposed method.
> > The table below shows the inference time with and without our method.
> >
> > || MM-diffusion | AudioLDM / AnimateDiff | Auffusion / VideoCrafter2 |
> > | :---: | :---: | :---: | :---: |
> > | w/o ours [sec/sample] | 1.09 | 3.10 | 4.45 |
> > | w/ ours [sec/sample] | 1.25 | 3.77 | 4.91 |
> > | Overhead | 15% | 22% | 10% |
> >
> > The inference time degrades around 10 to 20 \%, which is compatible with the typical classifier guidance [1] or is significantly better than classifier-free guidance [2] (it doubles the computation cost in the case of a single condition).
> > This overhead can be mitigated by designing a more computationally efficient network architecture.
> > We also include these results in Table 12 and the discussion above in appendix A.9.

---

> > > ### Author Response · Authors · 2024-11-25
> > > **Look forward to your responce**
> > >
> > > Thank you again for your constructive and insightful feedback. As the rebuttal period approaches its end, we are reaching out to confirm whether our responses have sufficiently addressed your concerns and questions. We would greatly appreciate it if we could have an active discussion regarding your feedback. Thank you, and we look forward to your response.

---

> > > > ### Comment · Reviewer_U5ht · 2024-11-26
> > > >
> > > > The authors have well addressed my concerns. I decide to maintain my score.

---

> > > > > ### Author Response · Authors · 2024-11-28
> > > > >
> > > > > Thank you for your response. We really appreciate your constructive feedback.

---

### Official Review · Reviewer_Kk8X · 2024-11-02

**Soundness:** 3
**Presentation:** 3
**Contribution:** 2
**Rating:** 6
**Confidence:** 4

**Summary:**

This paper aims to leverage the pre-trained single-modal diffusion models for generating aligned video and audio. To achieve this objective and involve minimal computing costs, the authors proposed a lightweight joint guidance module to adjust the independent distribution from base models to match the joint distribution over audio and video by designing and training a discriminator. Based on the experimental results on the benchmark datasets, it seems that the proposed method improves the alignment score of generated audio and video while involving reduced trainable parameters.

**Strengths:**

1. It is interesting to solve the joint audio-video generative problems by adjusting the pre-trained modality-specific generators with minimal trainable parameters rather than fully training the joint audio-video generation models.
2. Sufficient mathematic equations and proofs are given to support that the discriminator-based guidance can adjust the estimated scores from audio diffusion and video diffusion to approximately match the score of joint audio-video distribution.
3. The experimental results shown in the table present the proposed method can improve the alignment performance of generated audio and video in both in-domain and out-of-domain settings.
4. The paper is generally well-written.

**Weaknesses:**

1. The description that training a discriminator can adjust the pre-trained modality-specific diffusion models towards a joint audio-video generator is not very clear. For example, how does the discriminator-based guidance achieve both semantic and temporal alignment?  It is better to make it more clear.
2. The improvement of metric score is limited especially for the OOD setting. In addition, from Table 4, the FVD and IB-TV scores of AudioLDM/AnimateDiff with the proposed guidance module are even worse.
3. Based on the provided video files, although the performance in the IND is improved by the proposed method, the OOD setting performs poorly in generating high-quality video with the audio track.
4. It is meaningful to see how different base models affect the performance. Whether more powerful pre-trained video and audio generators further improve the performance when equipped with the same guidance module?

**Questions:**

1. Did you adopt other evaluation metrics to evaluate the proposed method on the performance of temporal alignment between generated audio and video? Because it seems that the ImageBind score is just for semantic alignment.
2. Did you try different discriminator structures or larger trainable parameters to see if there will be future performance improvement?
3. Why the metric scores of MM-Diffusion in Table 2 are different from the ones of the original paper?
4. It is better to compare the proposed method with "Seeing and Hearing" discussed in the related works by using the same pre-trained base models.

---

> ### Author Response · Authors · 2024-11-20
> **Rebuttal Response (1/3)**
>
> We appreciate the constructive feedback from the reviewer. We would like to answer the questions and the comments below.
>
> ## Explanation about adjusting pre-trained modality-specific diffusion models towards a joint generator by training a discriminator (Weakness 1)
> We thank the reviewer to point this out.
> We would like to clarify how our proposed method guides base models toward a joint generator in this response.
>
> Our proposed discriminator distinguishes between samples from an independent distribution $p(x)p(y)$ and ones from a joint distribution $p(x, y)$.
> Intuitively, the discriminator has knowledge about the difference between non-aligned samples and well-aligned ones.
> Using the outputs from this discriminator, we can shift the base models' independent generation processes toward a joint generation process.
>
> More specifically, we formulate our guidance to fill the gaps between the score function for a single-modal distribution and that for a joint distribution (Eqs. (7) and (8)).
> We show that these gaps can be estimated by a single discriminator trained for distinguishing between the samples generated independently by the base models and ones sampled from the real paired datasets (Eqs. (11) and (12)).
> We argue that the function of our discriminator is just shifting the score computed on a single-modal distribution toward that on a joint distribution, which is obtained by the dataset used for the training.
> In other words, we do not explicitly design our method to manipulate a specific type of alignment.
> The type of alignment mainly achieved in our model is determined by both the nature of the real dataset and the inductive bias of the joint discriminator.
> We train the discriminator on the ground truth audio-video pairs that are aligned both semantically and temporally.
> With its guidance, the generated samples can be considered sampled from the target joint distribution, resulting in higher semantic or temporal alignment.
>
> ## Limited improvement of the metric scores for the OOD setting (Weakness 2)
> As the reviewer pointed out, we observed slight performance degradation on only a few metrics: FVD on the VGGSound, IB-TV on the VGGSound and the Landscape datasets for AudioLDM/AnimateDiff.
>
> Before discussing these degradations in detail, we argue that our primary focus is the cross-modal alignment score, which measures how well our proposed method integrates base models into a joint generation model.
> Our proposed method successfully improves cross-modal scores for all settings with a par or even better performance on the single-modal fidelity score except the ones described above.
> This illustrates that our proposed method effectively shifts the single-modal scores to the joint scores while keeping (or even improving) the performance of each modality.
>
> Then, we would like to discuss the degradation in detail.
> One possibility causing this degradation is that, as we discussed in our manuscript, the captions added to the videos are biased to describe visual information, and the base audio models struggle to generate high-quality audio with these captions.
> To compensate for this bias, our model may focus more on aligning the score of audio to the target than the video.
> The IB-TA scores of AudioLDM / AnimateDiff (A) on both datasets are significantly worse than those of Auffusion / VideoCrafter2 (B), and performance improvement on IB-TA for (A) is more significant than that of (B) (15% and 8% improvements for (A) whereas 5% and 3% for (B) on the same datasets).
> This indicates that our discriminator focuses more on the audio part, resulting in a slight performance degradation in the IB-TV score.
> This bias would be mitigated by adopting a more sophisticated captioning strategy or discriminator's network architecture.
>
> ## The OOD setting performs poorly in generating high-quality video with the audio track (Weakness 3)
> Since we propose a method extending base models into a joint generation model, the quality of generated samples strongly depends on the performance of base models.
> Even though we adopted one of the state-of-the-art models publicly available (e.g., Auffusion and VideoCrafter2), the quality of these samples lags behind closed models, especially in the open domain generation (we used closed-domain datasets that include a less diverse landscape or dancing scene in the IND setting).
> However, our primary contribution is proposing a novel, computationally efficient way to integrate pre-trained single-modal models into a joint generation model.
> Our method can be applied to the new state-of-the-art models, as we verified its generalizability with various base models and datasets through experiments.
> Adopting new state-of-the-art models proposed in the future will improve performance under difficult settings, including OOD.

---

> ### Author Response · Authors · 2024-11-20
> **Rebuttal Response (2/3)**
>
> ## Different (more powerful) base models (Weakness 4)
> Compared to AudioLDM/AnimateDiff, Auffusion/VideoCrafter2 are stronger base models, and the results for these models (the bottom two rows in Tables 3 and 4) show the effectiveness of our proposed method even with more powerful base models.
> Even for those more powerful base models, our proposed method successfully improves both single-modal fidelity and cross-modal score, highlighting our method can be effective even in the challenging setting, in which the base models already achieve good generation results.
>
> ## Other evaluation metrics for temporal alignment between generated audio and video (Question 1)
> Thank you for your question.
> We computed the AV-Align score [1] to evaluate the temporal alignment between generated audio and video.
> The tables below show the results.
>
> | Landscape | AV-Align |
> | --- | --- |
> | AnimateDiff + AudioLDM | 0.319 |
> | AnimateDiff + AudioLDM + ours | 0.319 |
> | Auffusion + VideoCrafter2 | 0.289 |
> | Auffusion + VideoCrafter2 + ours | **0.304** |
>
> | VGGSound | AV-Align |
> | --- | --- |
> | AnimateDiff + AudioLDM | 0.387 |
> | AnimateDiff + AudioLDM + ours | **0.390** |
> | Auffusion + VideoCrafter2 | 0.363 |
> | Auffusion + VideoCrafter2 + ours | **0.374** |
>
> Our proposed method generates samples with better AV-align scores, indicating that our guidance successfully shifts the distribution of the generated samples towards the target joint distribution.
> While our proposed method is not explicitly designed to improve a specific type of alignment, it implicitly learns what the joint audio-video pairs should be from the dataset, resulting in better semantic and temporal alignment scores.
>
> Note that this metric compares the onsets in audio and the peaks of optical flow in video, measuring how the object movement and its corresponding sound align in temporal perspective.
> Although our proposed method improves the AV-Align scores, it might be unsuitable for evaluating ambient sound, such as the landscape dataset.
> How to evaluate the temporal alignment of sounding videos should be an open issue to be explored.
>
> [1] "Diverse and aligned audio-to-video generation via text-to-video model adaptation," AAAI 2024
>
> ## Different discriminator structures or larger trainable parameters (Question 2)
> Thank you for your question.
>
> > Larger trainable parameters
>
> We tested two settings for larger trainable parameters: scaling channel size or number of layers.
> These results are shown in Tables 8 and 9 in the appendix.
> In short, both settings slightly improve the cross-modal alignment and fidelity.
>
> > Different discriminator structures
>
> We additionally tested transformer-based structures for implementing our discriminator and compared them with our default discriminator structure.
>
> In short, we observed performance improvement of all metrics except FVD, which is on par by incorporating a transformer encoder to fuse the extracted audio and video features.
> The Table below shows the notable results from Table 10 in the updated manuscript (the details about the network architecture are explained in appendix A.8).
>
> | Architecture | FVD | IB-TV | FAD | IB-TA | IB-AV |
> | :---: | :---: | :---: | :---: | :---: | :---: |
> | Auffusion / VideoCrafter2 | 831 | 0.302 | 5.32 | 0.190 | 0.192 |
> | + Ours default | **704** | 0.302 | 4.79 | 0.197 | 0.201 |
> | Res / Naive transformer | 709 | **0.303** | **4.69** | 0.200 | 0.202 |
> | Res / Frieren | 710 | 0.302 | 4.77 | **0.202** | **0.205** |
>
> From this experiment, we show that a more sophisticated discriminator design would improve the performance of joint generation by our proposed method, which would be an interesting direction for future work.
> We updated our manuscript to include the results and discussion of this ablation study of the discriminator structure in appendix A.8 and Table 10.
> Please see them for more details.

---

> ### Author Response · Authors · 2024-11-20
> **Rebuttal Response (3/3)**
>
> ## Reason why metric scores of MM-Diffusion differ from the original papers (Question 3)
> Thank you for raising your concern about the mismatch in the metric scores.
> We generated pairs of audio and video by using the official implementation of MM-Diffusion and conducted evaluation on these samples.
> Although it is unclear that what causes the difference between our reproduced results and reported numbers in the original papers, this mismatch is also reported in one of the issues raised in their repository [1] and a previous work [2] (FVD=\~1141 and FAD=\~7.75 for Landscape in [2] vs. FVD =\~ 229 and FAD =\~ 9.39 in the MM-Diffusion paper).
> One clear difference is that we used implementations of FVD and FAD widely used in the community (see appendix A.3 for more details), whereas MM-diffusion uses their own implementation for computing these metrics.
> We consider that our evaluation is fair as we used the same implementation of evaluators to compare all methods.
>
> [1] https://github.com/researchmm/MM-Diffusion/issues/17
> [2] "Seeing and hearing: Open-domain visual-audio generation with domain visual-audio generation with diffusion latent aligners," CVPR 2024
>
> ## Comparison between "Seeing and Hearing" (Question 4)
> Since they do not release the code for joint video and audio generation in their official repository [1], we cannot directly compare their method and ours.
> Therefore, here, we compare our method with Seeing and Hearing (SaH) from the perspective of theoretical difference and clarify the benefit of our method by using the reported numbers in their paper.
>
> Our method differs from SaH by explicitly guiding base models' generation process toward a specific joint distribution.
> Specifically, we formulate our method to bridge the gap between an independent distribution $p(x)p(y)$ and a joint distribution $p(x, y)$.
> We assume that the training data is a sample of the target distribution $p(x, y)$ and derive the loss function to match the distribution learned by the base models with the target one.
> Therefore, the generation with our proposed method can be considered sampling from the target distribution $p(x, y)$.
> In contrast, SaH guides the generated samples to be closed in the latent space of a pre-trained foundation model, ImageBind.
> Although they successfully improve the audio and video alignment, the distribution of its generated samples is not explicitly defined.
> This difference makes our method preferable regarding two aspects: i) effectiveness as a guidance module for a joint generation model and ii) customization of base models for the specific dataset.
>
> From the perspective of i), our guidance can be applied to the joint generation model to boost its performance.
> As shown in Table 2, our method successfully improves both FAD and FVD of MM-Diffusion.
> In contrast, SaH struggles to improve both metrics of MM-Diffusion (FAD improves from 7.7 to 6.4 while FVD degrades from 1141 to 1174 as they reported in Table 1 in their paper).
> Our method provides a similar benefit to the classifier or classifier-free guidance commonly used in conditional generation.
>
> From the perspective of ii), our method successfully guides the same base models toward different joint distributions using several datasets (e.g., Landscape and VGGSound datasets).
> Tables 3 and 4 demonstrate this success as our method improves the FAD and FVD computed on each target dataset.
> Note that it is difficult to achieve this improvement by SaH because there is no guarantee that the distribution of generated samples by their method is closed to the target distribution.
> Our method allows us to customize base models for different datasets to achieve sampling from the specific target distribution.
>
> [1] https://github.com/yzxing87/Seeing-and-Hearing?tab=readme-ov-file#todo

---

> > ### Author Response · Authors · 2024-11-25
> > **Additional results for the Weakness 2 and 3 (consern about the performance in the OOD setting)**
> >
> > To address your concern about the performance of our method in the OOD setting, we additionally conducted a subjective evaluation of the samples generated by Auffusion/VideoCrafter2 with and without our method.
> > In this evaluation, we randomly sampled 10 labels from the VGGSound dataset.
> > We used them for the generation by both methods, resulting in 10 pairs of two videos (one is by base models, and the other is by base models with our method).
> > We asked the evaluators four questions for each pair of two videos (two videos are shown in random order): *"Which video is of higher quality regardless of audio?"*, *"Which audio is of higher quality regardless of video?"*, *"Which pair of audio and video has better semantic alignment?"*, and *"Which pair of audio and video has better temporal alignment?"*.
> > For each question, the evaluators can select the answer from three options: *"Video A is better"*, *"Video B is better"*, and *"N/A"*.
> > We collected 280 answers in total from 7 evaluators.
> > The following table shows the preference ratio of our method for each aspect:
> >
> > || Video Quality | Audio Quality | Semantic Alignment | Temporal Alignment |
> > | :---: | :---: | :---: | :---: | :---: |
> > | Auffusion / VideoCrafter2 | 4.76%  | 14.28% | 7.94%  | 9.52%  |
> > | Ours                      | **93.65%** | **49.21%** | **84.12%** | **80.96%** |
> > | N/A                       | 1.59%  | 36.51% | 7.94%  | 9.52%  |
> >
> > Although the number of evaluators is limited due to the time limitation, these results demonstrate that our method successfully improves the quality and alignment of audio and videos.
> > We hope these results support the effectiveness of our method and dispel your concern about the performance of our method.

---

> > > ### Author Response · Authors · 2024-11-25
> > > **Look forward to your response**
> > >
> > > Thank you again for your constructive and insightful feedback. As the rebuttal period approaches its end, we are reaching out to confirm whether our responses have sufficiently addressed your concerns and questions. We would greatly appreciate it if we could have an active discussion regarding your feedback. Thank you, and we look forward to your response.

---

> ### Comment · Reviewer_Kk8X · 2024-11-25
>
> Thank you very much for your response. Most of my concerns have been mitigated. However, I still have some questions:
> 1. As mentioned in the response to Weakness1, the proposed method is not designed for alignment and is based on the ground truth audio-video pairs that are aligned both semantically and temporally. But, during the sampling stage, how does the proposed method ensure to generate the audio and video semantically and temporally?
> 2. For the comparison with MM-Diffusion, can you try the metric implementation from MM-Diffusion on your proposed method to evaluate the generated performance?

---

> > ### Author Response · Authors · 2024-11-26
> >
> > Thank you for your response. We will answer your additional questions.
> >
> > > 1. As mentioned in the response to Weakness1, the proposed method is not designed for alignment and is based on the ground truth audio-video pairs that are aligned both semantically and temporally. But, during the sampling stage, how does the proposed method ensure to generate the audio and video semantically and temporally?
> >
> > Our proposed method ensures to generate the well-aligned audio and video by guiding the base models' generation process toward the joint distribution.
> > The guidance mechanism in our method is the same as the typical Classifier Guidance (C-guide) [1, 2], where we guide the noisy intermediate sample toward the target distribution at each generation step.
> > The key difference is that we use a binary classifier between a joint distribution (whose samples are well-aligned) and an independent distribution (whose samples are not aligned) as a guidance module, while the typical C-guide uses an image classifier that predicts a label of the image.
> > Please also refer to the pseudo-code of the inference of our proposed method shown in Algorithm 2 in our manuscript.
> >
> > We will clarify why guiding the generation process toward a joint distribution ensures audio and video alignment.
> > We consider that **sampling from a joint distribution is the most general way to ensure the alignment between audio and video**.
> > Since defining the alignment between audio and video is too complicated and challenging, we consider using a data-driven approach to achieve this alignment.
> > Specifically, we assume that the real pairs of audio and video (which follow the joint distribution of audio and video) have desired alignment properties.
> > If we can sample from this joint distribution, we can generate well-aligned audio and video.
> > From this assumption, **our goal is to build a generative model (or generation process), which models a joint distribution of video and audio**.
> > To achieve this, we directly bridge the gap between the base models' independent generation process (sampling from an independent distribution) and sampling from a joint distribution by the guidance of the discriminator classifying the samples from independent and joint distributions.
> > Note that, in inference, our discriminator takes noisy audio and video (with a timestep) as inputs and predicts the probability of how likely the current samples are from the joint distribution, and requires no real samples (the real samples are only used in the training).
> >
> > [1] "Score-based generative modeling through stochastic differential equations", ICLR 2021
> > [2] "Diffusion models beat gans on image synthesis", NeurIPS 2021
> >
> >
> > > 2. For the comparison with MM-Diffusion, can you try the metric implementation from MM-Diffusion on your proposed method to evaluate the generated performance?
> >
> > Yes. We run the metric implementation from MM-Diffusion on our method to evaluate the performance.
> > We used MM-Diffusion's official repository and followed its instructions for running its evaluation script (https://github.com/researchmm/MM-Diffusion?tab=readme-ov-file#test).
> > The following tables show the comparison between MM-Diffusion and MM-Diffusion + Our method using their metric implementation.
> >
> > | LandScape | FVD (MM-Diffusion) ↓ | KVD (MM-Diffusion) ↓ | FAD (MM-Diffusion) ↓ |
> > | :---: | :---: | :---: | :---: |
> > | MM-Diffusion | 351 | 20.68 | 11.11 |
> > | + Ours       | **301** | **17.28** | **10.56** |
> >
> > | AIST++ | FVD (MM-Diffusion) ↓ | KVD (MM-Diffusion) ↓ | FAD (MM-Diffusion) ↓ |
> > | :---: | :---: | :---: | :---: |
> > | MM-Diffusion | 610 | 32.71 | 9.47 |
> > | + Ours       | **542** | **28.99** | **8.95** |
> >
> > Our method successfully improves all metrics, even on the MM-Diffusion's metric implementation, similar to the reported numbers in our manuscript.
> > These results demonstrate the effectiveness of our method.
> > We hope these results mitigate your concern about the evaluation metric for comparing MM-Diffusion and our method.

---

> > > ### Comment · Reviewer_Kk8X · 2024-11-27
> > >
> > > Thanks for your response. I have some follow-up questions:
> > > 1. For the comparison to MM-Diffusion, why does your obtained score have a big difference from the one from the original paper, in terms of FAD? It is understood the reproduced score is a little different from the original paper. Did you use the official checkpoint to sample the videos and then apply the evaluation implementation?
> > > 2. As mentioned the paper mainly focuses on the cross-modal alignment by learning from the raw dataset. However, one main challenge of audio-video generation is to ensure a good temporal alignment between generated audio and video. I appreciate that the authors implement the AV-Align score on the baselines with or without the proposed method. It is better to show the alignment performance comparisons of your model with other existing models.
> > > 3. Apart from the alignment, the other challenge is to ensure the quality of each generated modality. From the demo (especially for the OOD setting), the visual quality seems to be poor. Maybe the potential reason is that the quality of each generated modality is based on the used modality-specific pre-trained. Can I understand that the proposed method or trainable module does not contribute to the quality of each generated modality?

---

> ### Author Response · Authors · 2024-11-28
> **Response to the follow-up questions (1/2)**
>
> Thank you for the follow-up questions. We will answer all the questions.
>
> > 1. For the comparison to MM-Diffusion, why does your obtained score have a big difference from the one from the original paper, in terms of FAD? It is understood the reproduced score is a little different from the original paper. Did you use the official checkpoint to sample the videos and then apply the evaluation implementation?
>
> Yes, we used the official checkpoints and the official implementation (the generation script) to sample the videos, and then we applied the official evaluation implementation.
> Regarding FAD, the reported number in the original paper is 9.39 in the Landscape dataset, but our obtained score is 11.11.
> We are still determining what causes this difference.
> Note that, as we mentioned in the rebuttal response above, similar differences are reported by the issue on the official repository [1] or the reproduced results in the previous work [2].
>
> [1] https://github.com/researchmm/MM-Diffusion/issues/17
> [2] "Seeing and hearing: Open-domain visual-audio generation with domain visual-audio generation with diffusion latent aligners," CVPR 2024
>
> > 2. As mentioned the paper mainly focuses on the cross-modal alignment by learning from the raw dataset. However, one main challenge of audio-video generation is to ensure a good temporal alignment between generated audio and video. I appreciate that the authors implement the AV-Align score on the baselines with or without the proposed method. It is better to show the alignment performance comparisons of your model with other existing models.
>
> Thank you for your suggestion.
> We computed the AV-Align score for the existing models to comprehensively compare our method with existing models regarding both semantic and temporal cross-modal alignment.
> We updated our manuscript to include the explanation and results of this experiment (in appendix A.10 and Table 13).
>
> During the evaluation of AV-Align, we noticed that our previous results were too high.
> We fixed this problem by adjusting the interval of onset detection to be compatible with TempoToken's configuration and computed AV-align scores for all models.
> Please refer to appendix A.10.1 in the updated manuscript for more details on the AV-Align computation.
> The table below compares our method with other existing models regarding cross-modal alignment.
>
> | VGGSound | AV-Align | IB-AV |
> | :---: | :---: | :---: |
> | Samples from the dataset (GT)               | 0.296 | 0.336 |
> | AnimateDiff -> SpecVQGAN                | 0.288 | 0.084 |
> | VideoCrafter2 -> SpecVQGAN              | 0.291 | 0.089 |
> | AnimateDiff -> Diff-Foley               | 0.295 | 0.098 |
> | VideoCrafter2 -> Diff-Foley             | 0.319 | 0.105 |
> | AnimateDiff + AudioLDM                  | 0.320 | 0.121 |
> | AnimateDiff + AudioLDM + ours           | **0.330** | 0.127 |
> | VideoCrafter2 + Auffusion               | 0.268 | 0.192 |
> | VideoCrafter2 + Auffusion + ours        | 0.287 | 0.201 |
> | VideoCrafter2 + Auffusion + ours (Res / Frieren) | 0.292 | **0.205** |
>
> For the AV-Align, "AnimateDiff + AudioLDM + ours" achieves the best score among all models.
> While it also achieves a better IB-AV score than existing models, the IB-AV score significantly lags behind GT's.
> Using more sophisticated base models (VideoCrafter2 + Auffusion + ours) significantly improves the IB-AV score, which has a comparable AV-Align score to the GT.
> Notably, our proposed method improves both AV-Align and IB-AV scores of both base models.
>
> We also observed that using a more sophisticated network structure ("VideoCrafter2 + Auffusion + ours (Res / Frieren)") improves both scores.
> How to design the discriminator to improve the specific types of alignment would be a possible direction of future work.
> Additionally, how to create a dataset would be a possible direction to improve the alignment.
> For example, Diff-Foley proposes using time-shifted audio and video pairs to train audio-visual latent space, which is used for the condition of their generative model, to improve the temporal alignment.
> In our experiments, we only used generated samples from base models for the fake samples.
> Adding such fake pairs as simulated temporal misalignment for the fake samples would improve the alignment of generated samples, which we leave for future work.
>
> We also included this discussion in the updated manuscript in appendix A.10.2.

---

> ### Author Response · Authors · 2024-11-28
> **Response to the follow-up questions (2/2)**
>
> > 3. Apart from the alignment, the other challenge is to ensure the quality of each generated modality. From the demo (especially for the OOD setting), the visual quality seems to be poor. Maybe the potential reason is that the quality of each generated modality is based on the used modality-specific pre-trained. Can I understand that the proposed method or trainable module does not contribute to the quality of each generated modality?
>
> We respectfully disagree with your understanding.
> Our proposed method (or trainable module) has the capacity to contribute to the quality of each generated modality within certain limits.
> The quantitative results (Tables 3 and 4) show that the FVD improves in three out of four settings, and the FAD improves in all four settings.
> These improvements in the FVD and FAD indicate that the proposed method successfully aligns the distribution of the generated samples with that of the real dataset.
> Since the quality of the real dataset should be high (or at least higher than the generated samples by the base models), the generated samples with our method have higher quality than those of base models, indicating our method's contribution to the quality of each generated modality.
> Moreover, the additional subjective evaluations we mentioned in the rebuttal response above also support the audio and video quality improvement, at least against the generated samples from the base models.
>
> On the other hand, we also notice the limitation of our proposed method.
> As you rightly pointed out, the quality of each generated modality strongly depends on the performance of the base models.
> Our proposed method guides the generated samples to have a higher density ratio between a joint distribution and independent distribution ($p(x, y) / p(x)p(y)$).
> Intuitively, our method allows us to pick the samples likely to be from a real joint dataset within (not far from) the range of base models' generation.
> We introduce a denoise loss to strengthen the guidance effect beyond the distribution gap in the OOD case.
> However, the base models still contribute a lot to the quality of the generated samples, especially when we try to guide the large base models by a lightweight module with a limited number of parameters.
> While a lightweight module struggles to improve the quality significantly, one of the benefits of the lightweight module is that we can easily apply our method to any new models released in the future.
> We expect the single modal generative models to continuously improve by introducing various sophisticated network designs, and combining new SoTA models with our proposed method (with less effort thanks to our model-agnostic formulation) will improve the quality of each generation modality.
>
> Apart from updating base models, one possible way to improve the quality of each modality is the curation of the training dataset.
> In the OOD setting, we used the LandScape or VGGSound dataset.
> The LandScape dataset is of high quality but on a small scale, while the VGGSound is large scale but not of high quality enough.
> Since our method successfully improves the FAD and FVD, using high-quality videos for the training would further improve the quality of each generated modality.

---

> > ### Author Response · Authors · 2024-12-02
> > **Look forward to your response**
> >
> > Thank you again for your invaluable and insightful feedback.
> > Based on your follow-up questions, we updated our manuscript to include the results of the temporal evaluation. We believe this provides a more comprehensive comparison of our method with other models as a joint audio-video generator.
> > We also clarified the reproduction of MM-Diffusion and the effect on the quality of each generated modality in the OOD setting.
> >
> > We hope that our responses have satisfactorily addressed your concerns, and we would appreciate it if you considered raising the score for our work.
> > If you have additional concerns or questions, we are more than willing to answer.
> > Thank you again for your valuable feedback, and we sincerely look forward to your response.

---

> > > ### Comment · Reviewer_Kk8X · 2024-12-02
> > >
> > > Thank you for your response. I'm willing to raise my rating to 6.

---

> > > > ### Author Response · Authors · 2024-12-03
> > > >
> > > > Thank you for your response. We really appreciate your constructive feedback.

---

### Official Review · Reviewer_F4ji · 2024-11-04

**Soundness:** 3
**Presentation:** 3
**Contribution:** 2
**Rating:** 6
**Confidence:** 3

**Summary:**

The submission presents a novel approach to audio-video joint generation with frozen single-modal generative models using a discriminator-guided cooperative diffusion process. The authors leverage existing pre-trained single-modal models for audio and video, aiming to reduce computational costs while enhancing multi-modal alignment. This approach, distinct from traditional model-specific or architecture-dependent techniques, incorporates a lightweight guidance module trained as a discriminator. The model effectively distinguishes between real and fake audio-video pairs, providing stability through a denoising regularization function inspired by standard diffusion methods.

**Strengths:**

* The proposed method is well motivated and clearly presented, with a focus on achieving joint audio-video generation with low computational cost.
* The proposed method is training-based but model-agnostic, which can be more broadly applicable than model-specific designs.
* Extensive results on multiple datasets demonstrate the effectiveness of the proposed method in improving modality alignment and generation quality of videos and audio.

**Weaknesses:**

* While the single-modal diffusion models remain frozen, the computation cost for training the joint discriminator, including forward passes through the frozen modules, is not reported. This cost needs to be compared to the total pretraining cost and a reasonable baseline of jointly trained diffusion models on both modalities.
* When text condition is provided to both diffusion models, the captions act as a weak alignment signal. Better experimental designs would be desired to remove the impact of this signal.

**Questions:**

* As mentioned in the Weaknesses, could the authors compare the training cost for different training setups?
* There are many values omitted in the FVD columns of Table 3 and 4. Are they the same as the corresponding rows below or different? Could the authors either explain or provide new values?
* Can the proposed method generalize to other pairs of modalities beyond video and audio?
* Will the proposed method remain valuable if diffusion models jointly trained on both modalities become widely available? If so, what does the cost-performance trade-off look like?

---

> ### Author Response · Authors · 2024-11-20
> **Rebuttal Response (1/2)**
>
> We appreciate the constructive feedback from the reviewer. We would like to answer the questions and the comments below.
>
> ## Computation cost for training joint discriminator (Weakness 1 & Question 1)
>
> Thank you for the question.
> In short, our proposed method can be trained around 1.5 to 2 times faster than baselines.
> We updated the manuscript to add the results of additional experiments about the computational cost for training and inference of our proposed method in appendix A.9, Tables 11 and 12.
> We measured the computation time for the forward and backward of the base models and our discriminator, and the times for fine-tuning base models without any extension and training of our method were compared.
> The summarized results are shown below.
>
> || MM-diffusion | AudioLDM / AnimateDiff | Auffusion / VideoCrafter2 |
> | :---: | :---: | :---: | :---: |
> | Baseline training 1step [sec] | 0.933 | 0.759 | 1.301 |
> | Ours training 1step [sec] | **0.462** | **0.521** | **0.616**|
> | Speed-up Ratio | 202% | 146% | 211% |
>
> Note that we include forward passes through the frozen modules in "Ours training 1step".
> These results concretely demonstrate the computational efficiency of our proposed method.
> Please see the updated manuscript for more details.
> The complete numbers for each part of the computation are shown in Table 11, and a detailed discussion is provided in appendix A.9.
>
> ## Text condition acts as a weak alignment signal (Weakness 2)
> Thank you for your suggestion.
> We wonder that experiments without text conditions are not practical because the performance of the base model in unconditional generation is too low.
> We consider that there are three levels of experimental designs in terms of the strength of text condition: i) unconditional generation, ii) weak conditional generation (like label), and iii) conditional generation with captions.
> However, in the case of i), when we completely removed the text conditions for the base models, the generated samples from the base models were significantly degraded.
> As reported in many articles[1, 2], diffusion models struggle to generate decent outputs without classifier-free guidance.
> On the other hand, we tested case ii) in the OOD setting experiments on the VGGSound dataset described in Section 4.3.
> We used the original labels of the VGGSound dataset, which are simple and do not contain any detailed information specifically describing each video instance (e.g., "dog barking" or "playing hokey").
>
> [1] High-Resolution Image Synthesis with Latent Diffusion Models, CVPR 2022
> [2] Photorealistic Text-to-Image Diffusion Models with Deep Language Understanding, NeurIPS 2022

---

> > ### Author Response · Authors · 2024-11-20
> > **Rebuttal Response (2/2)**
> >
> > ## Omitted values in the FVD columns of Tables 3 and 4 (Question 2)
> > This is because the FVD and IB-TV scores for the sequential approaches are the same as the corresponding results of the base models without our method.
> > For example, in Table 3, the FVD for both "AnimateDiff → SpecVQGAN" and "AnimateDiff → DiffFoley" is 852, and one for "VideoCrafter2 → SpecVQGAN" and "VideoCrafter2 → DiffFoley" is 700.
> > We added these numbers in Tables 3 and 4 with brackets.
> >
> > ## Generalizability to other pairs of modalities beyond video and audio (Question 3)
> > Thank you for pointing this out.
> > In theory, our method can be applied to any pair of modalities beyond video and audio.
> > We selected audiovisual joint data as our target application since this pair is one of the most popular multimodal data.
> >
> > Our method is general in terms of modalities, as we formulate that it reduces the gap between independent distribution $p(x)p(y)$ and joint distribution $p(x, y)$.
> > If we use a dataset for different pairs of modalities and different base models for these modalities, our method can be applied straightforwardly.
> > However, in practice, we should design the network architecture for the discriminator specific to the target modalities.
> > Although our formulation is general in terms of modalities, the types of alignment that would be achieved are not derived from our formulation but from the discriminator's inductive bias or the nature of the training dataset.
> > Since the optimal architecture would differ depending on the pairs of modalities, applying our method to different modality pairs is not trivial.
> > We leave the application to the different modalities for future work.
> >
> > ## Effectiveness of our methods on the joint generation models and cost-performance trade-off (Question 4)
> > In Section 4.2 (experiments under the IND setting), we verified that our method successfully boosts both the fidelity and alignment score of MM-diffusion, which is already trained on the joint dataset.
> > This indicates that our method can also be applied to the pre-trained joint generative models to further align generated samples to the target joint distribution, just like classifier guidance with the conditional generation model.
> > One important difference between classifier guidance and ours is that no training-based guidance method exists for joint generation models.
> > For example, there is a classifier-free guidance for the conditional generation, but it cannot be applied to the joint generation as it is because the conditions in joint generation are bidirectional.
> > In this sense, our method can be a baseline method for guiding a joint generation model to further improve the quality of generated samples.

---

> > > ### Author Response · Authors · 2024-11-25
> > > **Look forward to your response**
> > >
> > > Thank you again for your constructive and insightful feedback.
> > > As the rebuttal period approaches its end, we are reaching out to confirm whether our responses have sufficiently addressed your concerns and questions.
> > > We would greatly appreciate it if we could have an active discussion regarding your feedback.
> > > Thank you, and we look forward to your response.

---

> > > > ### Author Response · Authors · 2024-12-02
> > > >
> > > > As the extended rebuttal period is approaching its end, this is a gentle reminder to review our reply.
> > > > Based on your invaluable feedback on our work, we have updated our manuscript to include an experiment about the computation cost of our proposed method and added some numbers in Tables 3 and 4 for clarity.
> > > > We also clarified our intention of the experiment setting about the text condition and answered all your questions.
> > > >
> > > > We hope that our responses have satisfactorily addressed your concerns, and we would appreciate it if you considered raising the score for our work.
> > > > If you have additional concerns or questions, we are more than willing to answer.
> > > > Thank you again for your valuable feedback, and we sincerely look forward to your response.

---

> ### Author Response · Authors · 2024-12-03
> **Follow-up Before Discussion Period Ends**
>
> Dear Reviewer F4ji,
>
> With the discussion period ending in less than 8 hours, we would like to ensure we have satisfactorily addressed your concerns, as we believe we have.
> If there are any remaining questions or uncertainties, please feel free to leave a message before the rebuttal period ends.
> We would be happy to respond promptly.
>
> Thank you for your time and valuable feedback.
>
> Best regards,
> The Authors

---

> > ### Comment · Reviewer_F4ji · 2024-12-03
> >
> > Thank you for the additional results and explanations. My concerns have been mostly addressed and I'm raising my score to 6.

---

> > > ### Author Response · Authors · 2024-12-03
> > >
> > > Thank you for your response. We really appreciate your constructive feedback.

---

### Official Review · Reviewer_pfZF · 2024-11-06

**Soundness:** 3
**Presentation:** 2
**Contribution:** 3
**Rating:** 6
**Confidence:** 3

**Summary:**

This paper presents a method for constructing an audio-video generative model with low computational cost by leveraging pre-trained single-modal diffusion models for audio and video. The authors propose a lightweight joint guidance module that aligns audio and video outputs by adjusting scores to approximate the joint distribution, computed through the gradient of an optimal discriminator that distinguishes real and fake audio-video pairs.

**Strengths:**

1. This paper tackles an important problem in multi-modal generative modeling by introducing a cost-effective approach to synchronize audio and video generation, offering practical value for applications constrained by limited computational resources.
2. The custom loss function helps to steady the gradient of the discriminator. Extensive evaluations on benchmark datasets highlight enhanced fidelity for individual modalities and better alignment between audio and video, all achieved with minimal additional parameters, underscoring the efficiency of the proposed method.

**Weaknesses:**

1. Although I understand that the experiments are based on MM-Diffusion, the AIST and Landscape datasets are relatively small. The authors should discuss the generalizability of their method on larger datasets.
2. The paper claims that the method is model-agnostic; however, only one baseline is tested, which makes it difficult to confirm its general applicability.
3. Besides quantitative metrics, a subjective evaluation of the generated samples, would be beneficial to better assess the quality from a perceptual standpoint.

**Questions:**

1. How is the low computational cost demonstrated in this work? Are there specific experiments or data that quantify this claim?
2. While the quality of the generated 2-second samples is good, how does the model perform on longer video samples?

---

> ### Author Response · Authors · 2024-11-20
> **Rebuttal Response**
>
> We appreciate the constructive feedback from the reviewer. We would like to answer the questions and the comments below.
>
> ## Generalizability of our proposed method on larger datasets and application to the various base models (Weakness 1 & 2)
> While the reviewer rightly pointed out the relatively small size of the AIST++ and Landscape datasets, we would like to emphasize that we also conducted experiments on the VGGSound dataset.
> The VGGSound dataset is commonly used as a large-scale audiovisual dataset among publicly available ones, which contains around 200 times more videos than the Landscape and AIST++.
> This dataset serves as a robust validation of our method's applicability to larger datasets.
>
> It is also important to note that we tested our proposed method with not just one, but three base models, namely MMdiffusion, AudioLDM/AnimateDiff and Auffusion/VideoCrafter2.
> This demonstrates the versatility and applicability of our method across different base models.
> Please see Section 4.3 for the results of other two models; the results are shown in Tables 3 and 4 in our manuscript.
> The results of our experiments unequivocally demonstrate that our proposed method successfully enhances the cross-modal alignment score and video and audio fidelity for the various base models.
> This success is a clear testament to the generalizability and applicability of our method.
>
> ## Subjective evaluation (Weakness 3)
> Thank you for your suggestion.
> Our extensive quantitative evaluation of various base models and datasets illustrates the effectiveness of our proposed method, whose primary goal is to shift the score of the single modal distribution estimated by base models to be aligned with the score on the joint distribution.
> On the other hand, we agree that subjective evaluation would provide additional support for our proposed method's visual and aural improvements.
> We are planning to conduct a user study, but we are not sure whether we can collect enough feedback during this limited rebuttal period.
> If we successfully collect sufficient responses within the next week, we will provide additional subjective results for this rebuttal.
> Otherwise, we will continue to conduct a user study and include at least some results for the camera-ready version.
>
> ## Computation cost (Question 1)
>
> Thank you for the question.
> In short, our proposed method can be trained around 1.5 to 2 times faster than baselines.
> We updated the manuscript to add the results of additional experiments about the computational cost for training and inference of our proposed method in appendix A.9, Tables 11 and 12.
> We measured the computation time for the forward and backward of the base models and our discriminator, and the times for fine-tuning base models without any extension and training of our method were compared.
> The summarized results are shown below.
>
> || MM-diffusion | AudioLDM / AnimateDiff | Auffusion / VideoCrafter2 |
> | :---: | :---: | :---: | :---: |
> | Baseline training 1step [sec] | 0.933 | 0.759 | 1.301 |
> | Ours training 1step [sec] | **0.462** | **0.521** | **0.616**|
> | Speed-up Ratio | 202% | 146% | 211% |
>
> These results concretely demonstrate the computational efficiency of our proposed method.
> Please see the updated manuscript for more details.
> The complete numbers for each part of the computation are shown in Table 11, and a detailed discussion is provided in appendix A.9.
>
> ## Longer video generation (Question 2)
>
> As in typical diffusion-based generative models [1], our proposed method can generate longer video and audio by applying auto-regressive generation (i.e., the first several frames are fixed by the ones generated at the previous step, and the only remaining several frames are generated iteratively).
> We will conduct some experiments about this and hopefully provide results during this rebuttal period.
>
> [1] "Video Diffusion Models," ICLRW 2022

---

> > ### Comment · Reviewer_pfZF · 2024-11-25
> >
> > Thanks for your response. I'm willing to raise my rating to 6.

---

> > > ### Author Response · Authors · 2024-11-25
> > >
> > > Thank you for your response. We really appreciate your constructive feedback.

---

### Author Response · Authors · 2024-11-20
**To all reviewers**

First, we would like to express our gratitude to all the reviewers for taking their time and providing invaluable feedback on our manuscript.
We have carefully considered all the comments and questions.
Our detailed responses are posted in the reply to each review.

In response to the reviewers' comments, we have updated our manuscript.
We summarize our modifications as follows:

- We added Table 10 to show the performance improvement by using a more sophisticated network architecture (e.g., transformer-based architecture) for the discriminator.
- We added appendix A.8 to explain the details of network architecture and the experimental settings for Table 10 and discuss the results.
- We added Table 11 to show that the computational cost for training our proposed model is significantly lower than the baselines.
- We added Table 12 to clarify the computational overhead in inference introduced by our proposed method.
- We added appendix A.9 to explain the details of the experimental settings for Tables 11 and 12 and discuss these results.
- We added appendix A.10 and Table 13 to compare the temporal cross-modal alignment of our method with baselines.

Please see the corresponding part in the updated manuscript, as well as our responses to each review.
All the modifications are colored in red in the updated manuscript.

---

> ### Author Response · Authors · 2024-11-24
> **A kind reminder for discussion**
>
> Dear reviewers,
>
> We express our sincere gratitude for your valuable comments, which we consider crucial for improving our work. This is a gentle reminder to reviewers to review our replies. We are eager to engage in an active discussion. Thank you again for your attention and consideration.

---

### Meta-Review · Area_Chair_wjU5 · 2024-12-21

**Metareview:**

The paper introduces a novel approach for joint audio-video generation by leveraging classifier guidance to combine independent single-modal models. The proposed method enables synchronization across modalities while using off-the-shelf single-modal models as black boxes, without requiring additional fine-tuning. Empirical results across multiple models demonstrate the effectiveness of the approach.

Strength
* The paper addresses an important problem of joint audio-video generation and offering a novel, cost-effective solution.
* The proposed approach is model-agnostic, enabling its application to various pre-trained generative models without extensive re-training.
* The experimental results demonstrate the effectiveness of the method, particularly in improving cross-modal alignment.

Weakness raised in initial reviews
* The evaluation benchmarks are relatively small, raising concerns about generalizability to larger datasets.
* The claim of model-agnostic applicability requires stronger evidence.
* Lacks of subjective evaluation, where the authors claim it's ongoing but not fully presented during the rebuttal.
* The initial submission lacked details on the computational cost.
* Although the method improves alignment, the absolute performance of generated samples remains limited.
* The impact of the quality of the single-modal base models on the overall performance is not thoroughly analyzed.

This paper addresses an important problem in multi-modal generation and presents a reasonable, cost-effective solution. The empirical results validate the method’s effectiveness in aligning audio and video. While reviewers raised several concerns in their initial reviews, the rebuttal successfully addressed most of these issues. The additional results and clarifications provided during the rebuttal period substantially enhance the completeness of the paper. The paper is recommended for acceptance, provided the authors incorporate the rebuttal details into the final version.

**Additional Comments On Reviewer Discussion:**

The reviewers are satisfied with the rebuttal and either raised their scores or maintained their acceptance recommendations. The authors effectively addressed several concerns, including the limited benchmark size and the relatively low absolute performance, by providing detailed explanations and rationale. They also provide additional analysis of computational costs and results from a user study to resolve specific questions. Overall, the rebuttal and newly provided details sufficiently addressed the majority of the reviewers’ concerns.

---

### Decision · Program_Chairs · 2025-01-22

Accept (Poster)